# TraCeS: Learning Per-Timestep Constraint-Violation Credit from Sparse Trajectory-Level Labels

Siow Meng Low [1]   Ze Gong [2]   Akshat Kumar [1]

## Abstract

Ensuring safe behavior in reinforcement learning (RL) is challenging when safety constraints are implicit and cannot be densely measured. In many settings, supervision is limited to coarse approvals or rejections of whole trajectories (e.g., whether a rollout remained within an unknown safety threshold). We propose **TraCeS** (Trajectory-based Constraint Estimation for Safety), a method for learning **per-timestep violation credit** from such sparse trajectory-level labels. **TraCeS** trains a sequential violation estimator whose per-step credits factorize the predicted probability that a trajectory has **not yet violated** the constraint, and integrates this learned signal into constrained policy optimization. The method requires neither a known cost function nor a known threshold, and remains compatible with standard continuous-control algorithms. We provide a theoretical analysis of the approximation gap introduced by the learning objective, and demonstrate empirically that **TraCeS** improves constraint satisfaction and feedback efficiency over baselines across multiple continuous-control benchmarks, including long-horizon tasks and settings with noisy or inconsistent labels.

## 1. Introduction

Ensuring safe behavior in sequential decision-making remains challenging because safety constraints are often implicit, unmeasured, or unavailable during training (Krakovna et al., 2018; Liu et al., 2022; Malik et al., 2021; Saisubramanian et al., 2022). In safety-critical domains such as

robotics (LaValle, 2006), healthcare (Yu et al., 2021), and autonomous driving (Kiran et al., 2021), hard-coded constraints and hand-shaped cost functions are often brittle or fail to generalize. While reward shaping and cost penalties are common (Achiam et al., 2017; Ha et al., 2021; Stooke et al., 2020; Tessler et al., 2018), they may be difficult to specify and can miss *context-dependent acceptability criteria* (e.g., what constitutes a violation depends on operating conditions). Moreover, dense per-timestep supervision is frequently unavailable (Chirra et al., 2024), making safety alignment a bottleneck.

Consider autonomous driving (Appendix B.1). Avoiding a pothole or wildlife may require temporarily leaving the lane. Although quantities like *time off-lane* are measurable, mapping them to a fixed cost and threshold is brittle: the same deviation may be acceptable at low speed to avoid an obstacle, but unacceptable on a clear road at high speed or in poor weather. Thus both the *form* of a suitable cost signal and the *tolerance level* are context-dependent. In contrast, a human or monitor may still label whether a rollout remained within acceptable bounds.

Motivated by such settings, we adopt a *thresholded violation event* abstraction: a rollout is acceptable if an (unobserved) cumulative measure of undesirability stays below a tolerance, and unacceptable otherwise. This matches supervision as coarse trajectory accept/reject labels (or sparse checkpoints). Crucially, many safety violations are *monotone in time*: once a trajectory violates, it remains violated thereafter. This structure is consistent with cumulative-cost CMDP constraints (Altman, 1998; Ray et al., 2019), but here we do not assume access to the cost or the threshold.

Learning from coarse labels poses a credit assignment problem: labels indicate whether a violation occurred, but not *when* or *which* behaviors caused it. We propose **TraCeS**[1] (Trajectory-based Constraint Estimation for Safety), which learns *per-timestep constraint-violation credit* from sparse trajectory-level labels by training a sequential violation estimator whose per-step credits factorize the predicted probability that a trajectory has *not yet violated*

[1]School of Computing and Information Systems, Singapore Management University, Singapore [2]Shenzhen Institutes of Advanced Technology, Chinese Academy of Sciences, Shenzhen, China. Correspondence to: Siow Meng Low <smlow.2020@phdcs.smu.edu.sg>.

*Proceedings of the 43$^{rd}$ International Conference on Machine Learning*, Seoul, South Korea. PMLR 306, 2026. Copyright 2026 by the author(s).

[1]Code is available at https://github.com/siowmeng/TraCeS.

the constraint.

**Key advance.** This factorization turns delayed binary supervision into a dense, interpretable *per-step surrogate cost* that can be optimized with standard CMDP machinery, enabling constraint satisfaction *without* a hand-designed cost function or a known threshold. To reduce annotation burden under distribution shift, **TraCeS** additionally prioritizes which rollouts to label using an uncertainty criterion derived from the estimator.

For quantitative benchmarking, we use standard continuous-control tasks where an oracle cost exists to *construct* sparse accept/reject labels consistent with our supervision structure, while **TraCeS** never observes the oracle cost or threshold. Across 12 tasks from Safety Gymnasium, MuJoCo, and Bullet Safety Gym, **TraCeS** reliably learns constraint-satisfying policies under *zero knowledge* of cost and threshold, is substantially more label-efficient than a zero-knowledge baseline, and remains competitive with partial- and full-knowledge methods. Beyond aggregate performance, the learned credits yield localized per-timestep attributions aligned with violation events (Section 5.3). We also validate robustness in low-data regimes, under noisy labels, and in a small-scale human feedback study (Appendix C.6, C.11, C.9).

We make three primary contributions:

- **Problem formulation.** Safe policy learning from sparse trajectory-level labels under a monotone violation abstraction.

- **Method. TraCeS**: a sequential estimator for per-timestep violation credit, integrated with constrained policy optimization, plus a trajectory selection strategy to reduce labeling.

- **Theory and evaluation.** Approximation-gap analysis and empirical results on continuous-control tasks and retrospective feedback settings, including robustness to moderate-to-high label noise.

## 2. Related Work

**Safety in reinforcement learning.** Safe RL methods typically assume known per-step costs and a constraint threshold within a CMDP (Achiam et al., 2017; Stooke et al., 2020; Tessler et al., 2018). This includes constrained policy optimization (Achiam et al., 2017), Lagrangian approaches (Ha et al., 2021; Stooke et al., 2020; Tessler et al., 2018), and risk-sensitive RL (Chow et al., 2018). These assumptions can be limiting when safety is implicit or context-dependent; **TraCeS** instead learns per-timestep violation credit from sparse trajectory-level labels. Robust safe RL is largely orthogonal, focusing on perturbations or uncertainty rather than sparse supervision of implicit constraints (Liu et al., 2023b; Queeney & Benosman, 2023; Tessler et al., 2019).

**Runtime safety enforcement.** Shields and safety filters enforce safety at execution time by restricting actions given an explicit specification and suitable abstractions/models (Alshiekh et al., 2018; Jansen et al., 2020). Control barrier functions similarly enforce invariance of a specified safe set, often via optimization-based filtering with known or reliably modeled dynamics (Ames et al., 2019). **TraCeS** is complementary: it targets settings where such specifications are unavailable and only sparse accept/reject feedback can be obtained.

**Inverse constraints and constraint learning.** Constraint learning has been studied from expert demonstrations (Liu et al., 2023a; Malik et al., 2021; Scobee & Sastry, 2020) or dense feedback (Saisubramanian et al., 2022), typically assuming either safe demonstrations or structured per-step supervision. RLSF (Chirra et al., 2024) learns safety costs from trajectory-level feedback but assumes knowledge of the threshold and number of constraints. In contrast, **TraCeS** operates without demonstrations, dense labels, or known constraint parameters.

**Learning from sparse trajectory-level feedback.** Preference-based RL learns reward models from trajectory-level comparisons (Christiano et al., 2017; Ibarz et al., 2018; Lee et al., 2021) or rankings (Brown et al., 2019; Wirth et al., 2017), and RLHF applies similar ideas in language domains (Bai et al., 2022; Ouyang et al., 2022; Rafailov et al., 2023). These primarily target scalar reward recovery, whereas **TraCeS** targets per-timestep *violation credit* for constraint satisfaction under a monotone not-yet-violated label structure, enabling CMDP-style optimization from sparse supervision.

**Return decomposition and delayed credit assignment.** **TraCeS** is also related to return-decomposition methods for delayed credit assignment, such as RUDDER (Arjona-Medina et al., 2019) and randomized return decomposition (RRD) (Ren et al., 2022). These methods redistribute delayed scalar returns into denser per-step reward signals to improve reward-based policy learning. In contrast, **TraCeS** does not aim to uniquely recover an oracle per-step cost from a scalar endpoint return. It learns a supervision-aligned surrogate violation credit from sparse binary non-violation labels under a monotone safety structure, and uses this surrogate as a constraint signal for policy optimization. Thus, the learned credits should be viewed as control-relevant safety surrogates rather than identifiable recovery of hidden ground-truth costs.

## 3. Background and Problem Setting

### 3.1. CMDPs as a benchmarking formalism

A Constrained Markov Decision Process (CMDP) (Altman, 1998) augments an MDP with a (typically nonnegative) cost and a constraint threshold. We write a CMDP as $(S, A, P, R, C, b, \gamma)$, where $S$ and $A$ are state and action spaces, $P(s' \mid s, a)$ is the transition kernel, $R(s, a)$ is reward, $C(s, a) \geq 0$ is a (latent) cost, $b$ is a constraint threshold, and $\gamma \in [0, 1]$ is a discount factor. A trajectory is $\tau = \langle s_0, a_0, \ldots, s_{T-1}, a_{T-1} \rangle$. Standard CMDPs optimize a parameterized policy $\pi_\theta$ via

$$\max_\theta \ \mathbb{E}_{\tau \sim \pi_\theta} \left[ \sum_{t \geq 0} \gamma^t R(s_t, a_t) \right]$$

$$\text{s.t.} \ \mathbb{E}_{\tau \sim \pi_\theta} \left[ \sum_{t \geq 0} \gamma^t C(s_t, a_t) \right] \leq b. \qquad (1)$$

In practice, episodes are truncated at a finite horizon $T$; we use the discounted form for consistency with common implementations.

**Our setting (implicit constraints).** Unlike standard safe RL, we assume the agent does *not* observe $C$ or $b$. Instead, supervision comes only as sparse binary labels indicating whether a trajectory/prefix has *not yet violated* an implicit constraint. We focus on the common *monotone* case where violations are irreversible (formalized in Assumption 3.1), consistent with thresholded cumulative constraints and nonnegative-cost benchmarks (Liu et al., 2022; Ji et al., 2023; Sootla et al., 2022).

### 3.2. Trajectory-level and prefix binary feedback

We assume supervision only as sparse binary labels over trajectories or prefixes. Let $\Psi(\tau_{0:t}) \in \{0, 1\}$ be an *unknown* labeling rule over prefixes $\tau_{0:t} = \langle s_0, a_0, \ldots, s_t, a_t \rangle$, where $\Psi(\tau_{0:t}) = 1$ means "not yet violated by time $t$" and $\Psi(\tau_{0:t}) = 0$ means "violated by $t$".

**Assumption 3.1** (Monotone violation / irreversibility). For any trajectory prefix $\tau_{0:t}$ and any $t' \geq t$, if $\Psi(\tau_{0:t}) = 0$ then $\Psi(\tau_{0:t'}) = 0$. Equivalently, once a violation has occurred, it cannot be "undone" by later actions.

Under Assumption 3.1, $\Psi(\tau_{0:t})$ is non-increasing in $t$ (once it flips to 0, it stays 0). A labeled prefix indicates whether a violation has occurred by that point, but not the first violating timestep: if the prefix up to step 20 is non-violating and the prefix up to step 40 is violating, then the first violation occurred somewhere between steps 21 and 40.

We observe a dataset of sparsely labeled prefixes $\mathcal{D} = \{(\tau_{0:t_i}^i, \psi^i)\}_{i=1}^N$ with $\psi^i = \Psi(\tau_{0:t_i}^i)$ and variable $t_i$ (e.g.,

one terminal label per episode or a few checkpoint labels). This supervision is sparse and delayed: it does not identify which steps caused the violation.

**Goal.** From $\mathcal{D}$, **TraCeS** learns a per-timestep violation-credit signal aligned with $\Psi(\cdot)$ and uses it for constrained policy optimization. Labeled prefixes need not be expert data; they may have either label $\psi \in \{0, 1\}$.

**Evaluation protocol.** For benchmarking against CMDP baselines, we use simulators where an oracle cost exists to *construct* labels matching the above supervision. However, **TraCeS** never uses oracle costs or the threshold during training.

## 4. TraCeS: Trajectory-based Constraint Estimation for Safety

**TraCeS** alternates between (i) training a *violation estimator* from sparse labels, (ii) improving the policy using the learned per-step violation credit, and (iii) selecting informative trajectories for labeling (Fig. 1).

Figure 2 illustrates the architecture of the **TraCeS** violation estimator. We begin by introducing the per-timestep violation credit formulation that underpins our estimator design.

### 4.1. Violation Estimator Learning

#### 4.1.1. VIOLATION CREDIT ASSIGNMENT

Consider an observed trajectory $\tau_{0:T-1}$, meaning a specific, realized sequence of states and actions $(s_0, a_0, s_1, a_1, \ldots, s_{T-1}, a_{T-1})$ collected from the environment. Each observed trajectory is labeled with a single binary label. The **TraCeS** violation estimator learns to decompose this sparse feedback into per-step violation credits.

**Definition 4.1** (Non-violation score of an observed prefix). For an observed prefix $\tau_{0:t}$, the estimator outputs $\hat{P}(\psi_{0:t} = 1 \mid \tau_{0:t})$, a predicted probability (score) that the constraint has not yet been violated by time $t$.

**Proposition 4.2** (Per-step violation credit). *The non-violation probability $\hat{P}(\psi_{0:T-1} = 1 \mid \tau_{0:T-1})$ can be factorized into per-step violation credits:*

$$\hat{P}(\psi_{0:T-1} = 1 \mid \tau_{0:T-1}) := \prod_{t=0}^{T-1} \hat{P}_t^\Delta. \qquad (2)$$

*Proof.* By the chain of ratios over nested prefixes,

$$\hat{P}(\psi_{0:T-1} = 1 \mid \tau_{0:T-1})$$
$$= \frac{\hat{P}(\psi_{0:0} = 1 \mid \tau_{0:0})}{1} \prod_{t=1}^{T-1} \frac{\hat{P}(\psi_{0:t} = 1 \mid \tau_{0:t})}{\hat{P}(\psi_{0:t-1} = 1 \mid \tau_{0:t-1})}, \qquad (3)$$

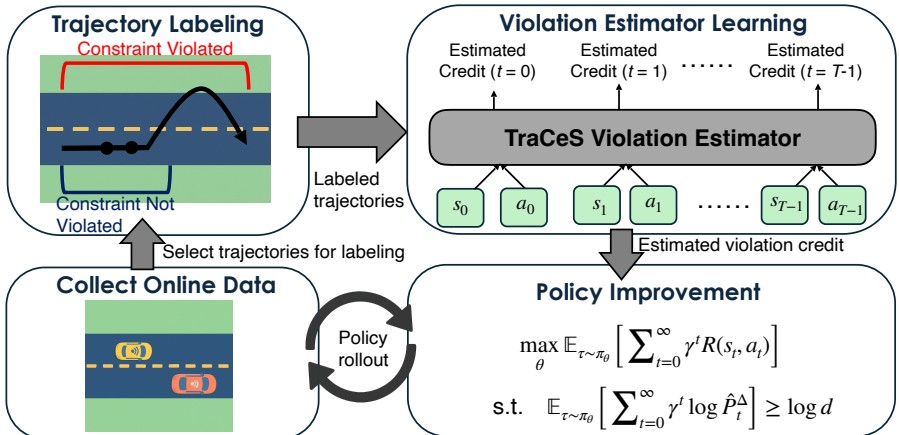

*Figure 1.* **TraCeS pipeline**. A selected subset of collected trajectories are labeled at the trajectory level. A violation estimator learns to assign per-timestep violation credit from these coarse labels, which is then used to optimize a policy. The resulting policy is used to collect new rollouts, forming an iterative learning loop.

and the product telescopes. Defining $\hat{P}_0^\Delta := \hat{P}(\psi_{0:0} = 1 \mid \tau_{0:0})$ and $\hat{P}_t^\Delta := \hat{P}(\psi_{0:t} = 1 \mid \tau_{0:t})/\hat{P}(\psi_{0:t-1} = 1 \mid \tau_{0:t-1})$ yields Eq. (2). □

**Change-factor view.** $\hat{P}_t^\Delta$ is a multiplicative *change factor* in predicted non-violation when extending an observed prefix by one step. In particular, $\hat{P}_t^\Delta \approx 1$ means step $t$ causes *little additional decrease* beyond $\tau_{0:t-1}$ (even if the prefix score is already small), whereas $\hat{P}_t^\Delta \ll 1$ flags a step that induces a sharp localized drop.

*Remark* 4.3 (Range and scope). Assumption 3.1 implies that the *true* probability of being unviolated cannot increase as a trajectory prefix is extended. Our estimator defines prefix scores via the multiplicative model $\hat{P}(\psi_{0:t} = 1 \mid \tau_{0:t}) := \prod_{k=0}^t \hat{P}_k^\Delta$ and enforces $\hat{P}_k^\Delta \in [0,1]$ by construction (negative log-normal credits; Section 4.1.2). Therefore, under our model,

$$\hat{P}(\psi_{0:t} = 1 \mid \tau_{0:t}) = \hat{P}(\psi_{0:t-1} = 1 \mid \tau_{0:t-1}) \cdot \hat{P}_t^\Delta \qquad (4)$$
$$\leq \hat{P}(\psi_{0:t-1} = 1 \mid \tau_{0:t-1})$$

i.e., extending a prefix cannot increase the predicted non-violation score. This matches settings where violations are irreversible (once violated, later actions cannot undo it). All quantities above apply to a *specific observed* prefix $\tau_{0:t}$ (a trajectory instance), not a policy-level expectation.

**Interpreting $\hat{P}_t^\Delta$ as violation credit.** $\hat{P}_t^\Delta$ measures the *incremental drop* in predicted non-violation when extending an observed prefix by one step. Thus, $\hat{P}_t^\Delta \approx 1$ assigns little additional evidence of violation to $(s_t, a_t)$ beyond what was already present in $\tau_{0:t-1}$ (even if the prefix is already likely violated), whereas $\hat{P}_t^\Delta \ll 1$ localizes steps that induce a sharp decrease, identifying salient contributors under the unknown rule $\Psi$. Crucially, because $\hat{P}_t^\Delta$ is an *increment*,

a step can occur inside a violating trajectory yet still receive $\hat{P}_t^\Delta \approx 1$ if similar steps also commonly appear in accepted trajectories (hence providing little discriminatory evidence of violation). Conversely, steps that are *diagnostic* of violation (i.e., disproportionately associated with rejected trajectories) tend to receive $\hat{P}_t^\Delta \ll 1$.

### 4.1.2. MODEL ARCHITECTURE

Proposition 4.2 highlights that $\hat{P}_t^\Delta$ depends on the ratio between the non-violation probabilities of prefixes $\tau_{0:t-1}$ and $\tau_{0:t}$. We therefore design the estimator (Figure 2) to maintain a summary representation $h_t$, which summarizes the constraint-relevant features of prefix $\tau_{0:t-1}$ and evolves over time, enabling localized estimation of per-step violation credit. We stress that this architecture estimates violation for an *observed* trajectory segment and the total timestep $T$ can vary across *observed* segments.

The shared encoder $f_w$ is a neural network that receives the current state-action pair $(s_t, a_t)$ and the previous summary vector $h_t$, and produces the updated summary vector $h_{t+1}$. The shared decoder module $g_w$ compares $h_t$ and $h_{t+1}$ to estimate the violation credit $\log \hat{P}_t^\Delta$ (in log form) at timestep $t$. Note that instead of producing the violation credit directly, the decoder outputs parameters $(\mu_t, \sigma_t)$ of a log-normal distribution, from which we sample $U_t \sim \text{Lognormal}(\mu_t, \sigma_t)$ and set $\log \hat{P}_t^\Delta := -U_t \leq 0$. This design yields a per-step uncertainty proxy (via the predicted variance of $\log \hat{P}_t^\Delta$), which we use for trajectory selection (Section 4.3). Summing these log-violation credits along a labeled prefix yields the log non-violation probability in Eq. (2) (in log form), allowing the estimator to be trained from binary labels on trajectories or labeled prefixes.

We emphasize that negative lognormal distribution is deliberately chosen to satisfy $0 \leq \hat{P}_t^\Delta \leq 1$. We clamp

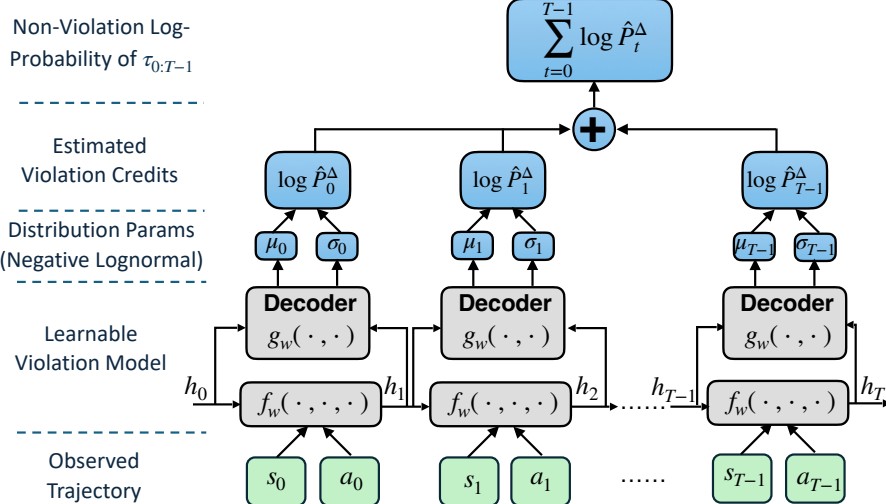

*Figure 2.* **TraCeS violation estimator**. The shared encoder $f_w$ processes each state-action pair $(s_t, a_t)$ along with the previous summary vector $h_t$ to produce summary vector $h_{t+1}$. A shared decoder $g_w$ maps $(h_t, h_{t+1})$ to a distribution over log-violation credits constrained to be non-positive (negative log-normal). The trajectory-level non-violation log-probability is approximated by summing these sampled scores in log space, leveraging the decomposition in Eq. (2). This design allows **TraCeS** to efficiently train on both short and long labeled prefixes, improving estimation in long-horizon settings.

$\log \hat{P}_t^\Delta \geq -7$ for numerical stability when summing log-credits over long horizons (equivalently, multiplying many small probabilities), which can otherwise lead to underflow and unstable gradients. Such clipping of log-probabilities is a standard stability device in deep learning implementations. We include a sensitivity study over the clamp value in Appendix C.12. Finally, a single long trajectory can provide multiple labeled prefixes (e.g., at checkpoints), increasing the amount of supervision and helping the estimator localize where the predicted non-violation probability drops in long-horizon settings.

### 4.1.3. LEARNING VIOLATION CREDIT

The estimated log non-violation probability of an observed labeled prefix, $\log \hat{P}(\psi_{0:t} = 1 \mid \tau_{0:t})$, is computed by summing per-step log-credits $\log \hat{P}_k^\Delta$. We train the estimator using binary cross entropy loss (Bishop, 2006; Goodfellow et al., 2016), which corresponds to maximum-likelihood estimation under a Bernoulli label model. Unlike RLSF (Chirra et al., 2024), which uses a surrogate objective, this directly fits $\hat{P}(\psi = 1 \mid \tau)$ to the observed labels to approximate the unknown labeling function $\Psi(\cdot)$.

### 4.2. Policy Improvement

The violation estimator produces per-step credits for an *observed* trajectory. We now show how to use these credits for safe policy optimization when the underlying cost function $C$ and threshold $b$ are unknown.

**From cost thresholds to trajectory-level acceptability.** In standard CMDP benchmarking, safety is specified by an (unknown to the agent) cumulative-cost threshold, e.g., $\Psi(\tau) = \mathbb{1}\{\sum_t C(s_t, a_t) \leq b\}$. This induces a binary accept/reject signal over trajectories without revealing the underlying cost. We therefore enforce safety directly in terms of the (unknown) acceptability function $\Psi$ by requiring that a policy produces acceptable trajectories with probability at least $d$:

$$\mathbb{E}_{\tau \sim \pi_\theta}[\Psi(\tau)] \geq d, \quad 0 \leq d \leq 1. \tag{5}$$

Eq. (5) is a chance-style constraint (satisfaction probability) induced by the *unknown* rule $\Psi$; we use it because $\Psi$ is exactly what supervision provides when $C$ and $b$ are unobserved. Since $\Psi(\tau) \in \{0, 1\}$, $\mathbb{E}_{\tau \sim \pi_\theta}[\Psi(\tau)]$ is exactly the probability (rate) that a trajectory sampled from $\pi_\theta$ is labeled acceptable. Although Eq. (5) is compatible with arbitrary implicit labeling rules $\Psi$, we instantiate $\Psi$ via cumulative-cost thresholding only for benchmarking against CMDP baselines; **TraCeS** itself does not assume this form.

The parameter $d$ specifies a desired acceptability rate (e.g., $d = 0.9$ means at least 90% of trajectories should be labeled acceptable). Unlike the hidden threshold $b$, whose scale depends on the unknown cost definition, horizon, and environment dynamics, $d \in [0, 1]$ is a scale-free policy-level requirement: $b$ defines the accept/reject boundary encoded by $\Psi$, while $d$ specifies how often the policy should satisfy it. Empirically, we set $d$ to 0.9 and conduct a sensitivity analysis of this hyperparameter (Appendix C.10).

Now we reformulate the constraint using the estimated violation credits. The following approximation result relies

on the structural monotonicity in Assumption 3.1, sufficient capacity of the estimator class to approximate the on-policy accept/reject rule, and small optimization/generalization error on the trajectory distribution induced by the current policy.

**Lemma 4.4.** *For a trajectory $\tau_{0:T-1}$ of length $T$, let $\hat{p}_\tau := \hat{P}(\psi_{0:T-1} = 1 \mid \tau_{0:T-1})$ denote the estimator's predicted non-violation probability. By construction, $\hat{p}_\tau = \prod_{t=0}^{T-1} \hat{P}_t^\Delta$. If $\hat{p}_\tau$ is a good proxy for the acceptability indicator $\Psi(\tau_{0:T-1})$ on the trajectory distribution induced by $\pi_\theta$, then the constraint (5) is approximated by $\mathbb{E}_{\tau \sim \pi_\theta}\left[\prod_{t=0}^{T-1} \hat{P}_t^\Delta\right] \geq d$.*

*Proof.* The factorization $\hat{p}_\tau = \prod_{t=0}^{T-1} \hat{P}_t^\Delta$ follows from Proposition 4.2. Training with binary cross entropy is maximum-likelihood estimation under a Bernoulli label model and is a *proper scoring rule*: in the population limit, its minimizer is the true conditional probability $\mathbb{P}(\Psi(\tau) = 1 \mid \tau)$ (under standard realizability/capacity assumptions). Thus, when the estimator is well-trained on the on-policy trajectory distribution, $\hat{p}_\tau$ provides a proxy estimate for $\Psi(\tau)$, yielding $\mathbb{E}[\Psi(\tau)] \approx \mathbb{E}[\hat{p}_\tau] = \mathbb{E}[\prod_t \hat{P}_t^\Delta]$. □

### 4.2.1. CMDP REFORMULATION

With Lemma 4.4, we can rewrite the constraint: $\log \mathbb{E}_{\tau \sim \pi_\theta}\left[\prod_{t=0}^{T-1} \hat{P}_t^\Delta\right] \geq \log d$.

The constraint is not in the standard CMDP form as the quantity inside the expectation is not a simple addition of violation credits. To address this in a principled manner, we derive a tractable lower bound using Jensen's inequality:

$$\log \mathbb{E}_{\tau \sim \pi_\theta}\left[\prod_{t=0}^{T-1} \hat{P}_t^\Delta\right] \geq \mathbb{E}_{\tau \sim \pi_\theta}\left[\sum_{t=0}^{T-1} \log \hat{P}_t^\Delta\right] \geq \log d\,. \tag{6}$$

By constraining the lower bound expression in Eq. (6) to be greater or equal to $\log d$, the original constraint is also satisfied. Now define the per-step surrogate cost $\tilde{C}_t := -\log \hat{P}_t^\Delta \geq 0$ and surrogate threshold $\tilde{b} := -\log d \geq 0$. Then the Jensen lower bound constraint $\mathbb{E}\left[\sum_{t=0}^{T-1} \log \hat{P}_t^\Delta\right] \geq \log d$ has exactly the same form as the standard CMDP constraint:

$$\mathbb{E}_{\tau \sim \pi_\theta}\left[\sum_{t=0}^{T-1} \tilde{C}_t\right] \leq \tilde{b}\,.$$

For continuing tasks, we use the discounted surrogate cost $\sum_{t \geq 0} \gamma^t \tilde{C}_t$ with $\gamma < 1$ for consistency with standard implementations. In practice, we apply the estimator recurrently along each rollout to produce $\hat{P}_t^\Delta$ at each step, so the surrogate cost can be evaluated on trajectories of varying length.

The CMDP constraint is now in standard form where the quantity inside expectation is a simple sum across timesteps, enabling direct application of constrained RL methods using the per-step surrogate cost $\tilde{C}_t$ estimated by **TraCeS**. Because **TraCeS** produces a per-step surrogate cost $\tilde{C}_t$, it can be combined with any standard CMDP solver; we instantiate it with PPO-Lagrangian (Ray et al., 2019) in our experiments (implementation details in Appendix B.3).

We now derive a bound on the approximation gap introduced in Eq. 6 when utilizing Jensen's inequality.

**Theorem 4.5** (Jensen-Gap Bound). *Suppose $\hat{p}_\tau \in [\epsilon, 1]$ for all trajectories $\tau$, where $\hat{p}_\tau := \hat{P}(\psi = 1 \mid \tau)$, and $\epsilon > 0$ is ensured by clamping log-credits to be bounded below. Then,*

$$\log\left(\mathbb{E}_{\tau \sim \pi_\theta}[\hat{p}_\tau]\right) - \mathbb{E}_{\tau \sim \pi_\theta}[\log \hat{p}_\tau] \leq \frac{1}{2\epsilon^2} \cdot \mathrm{Var}_{\tau \sim \pi_\theta}(\hat{p}_\tau)\,.$$

*Proof.* Refer to Appendix A.1. □

*Remark* 4.6. As training progresses, the estimator typically becomes more stable on the on-policy trajectory distribution, so $\mathrm{Var}(\hat{p}_\tau)$ decreases; consequently the Jensen gap shrinks.

**Tightness of the bound.** Theorem 4.5 is a worst-case diagnostic bound rather than a numerically tight certificate for all horizon lengths. With per-step clamping $\log \hat{P}_t^\Delta \geq -7$ over a segment of length $T$, the trajectory-level score satisfies $\hat{p}_\tau \geq \exp(-7T)$, so the lower bound $\epsilon$ can become small for long undiscounted segments. This can make the worst-case bound loose, although the looseness is conservative: a smaller $\epsilon$ enlarges the upper bound on the Jensen gap rather than making the surrogate constraint overly optimistic. Empirically, **TraCeS** remains stable on horizon-1000 tasks and in the clamp-sensitivity study in Appendix C.12.

### 4.3. Selecting Trajectories for Labeling

Figure 1 illustrates an iterative loop in which the estimator is periodically retrained using newly labeled trajectories. This is important because the trajectory distribution shifts as the policy changes. To reduce annotation burden, **TraCeS** prioritizes trajectories with high dispersion in the estimator's predicted surrogate costs.

We use coefficient of variation (CV), a normalized dispersion measure (Hima Bindu et al., 2019; Hastie et al., 2009), as a lightweight uncertainty proxy to select trajectories for labeling; this is a dispersion-based score rather than an ensemble-style epistemic uncertainty estimate.

Let $w$ denote the current estimator parameters and let $\tau$ be a fixed observed trajectory. At each step $t$, the estimator deterministically computes $(\mu_t, \sigma_t)$ from $(\tau, w)$ and samples a log-credit $Z_t \sim -\mathrm{Lognormal}(\mu_t, \sigma_t)$ (so $Z_t \leq 0$); define the sampled surrogate cost $\tilde{C}_t := -Z_t \geq 0$.

*Table 1.* Safety gymnasium and MuJoCo performance (eval environment) – part 1. Non-violating policies are shown in black bold; the best among non-violating methods is highlighted in blue. Violating policies are shown in gray. Standard deviations shown in parentheses.

| Task | Metric | PPO-Lagrangian (Oracle) | CPPO-PID (Oracle) | RLSF (Partial Knowledge) | *C-T* Baseline (Zero Knowledge) | TraCeS (Ours) (Zero Knowledge) |
|---|---|---|---|---|---|---|
| PointCircle1 | Reward | **46.6 (1.4)** | 46.8 (1.72) | **41.9 (1.8)** | 40.2 (1.1) | **43.2 (1.5)** |
| | Cost | **21.5 (7.2)** | 29.4 (4.9) | **1.6 (2.2)** | 25.3 (15.9) | **16.2 (7.2)** |
| | Labeled Trajectories | **NA** | NA | 3244 (966) | 12000 (0) | 5830 (918) |
| PointCircle2 | Reward | 41.7 (0.9) | **41.3 (0.8)** | **39.9 (0.8)** | **40.2 (0.8)** | **41.1 (0.9)** |
| | Cost | 29.1 (6.00) | **22.8 (7.3)** | **2.6 (4.5)** | **22.7 (13.5)** | **19.8 (9.5)** |
| | Labeled Trajectories | NA | **NA** | 3422 (589) | 12000 (0) | 5465 (1328) |
| CarCircle1 | Reward | **18.3 (0.4)** | **17.9 (0.3)** | 14.3 (2.2) | 18.0 (0.5) | **17.4 (0.9)** |
| | Cost | **23.5 (5.9)** | **23.7 (4.4)** | 33.8 (41.7) | 28.2 (11.7) | **11.3 (8.0)** |
| | Labeled Trajectories | **NA** | **NA** | 7479 (645) | 12000 (0) | 4005 (7) |
| CarCircle2 | Reward | 16.2 (0.2) | **14.9 (1.4)** | **13.7 (1.1)** | 15.3 (0.9) | **15.2 (0.9)** |
| | Cost | 28.3 (5.2) | **22.6 (4.3)** | **20.2 (17.0)** | 44.0 (24.0) | **11.9 (10.0)** |
| | Labeled Trajectories | **NA** | **NA** | 7601 (597) | 12000 (0) | **4341 (243)** |
| Ant | Reward | **3313.1 (53.8)** | 3284.7 (48.0) | 2220.7 (91.8) | 3096.7 (63.7) | 2885.4 (12.3) |
| | Cost | **14.4 (6.8)** | 12.8 (4.5) | 5.8 (2.6) | 20.3 (4.1) | 19.5 (5.3) |
| | Labeled Trajectories | **NA** | NA | 9036 (1289) | 6000 (0) | 4657 (551) |
| HalfCheetah | Reward | **3008.3 (52.9)** | 3007.9 (44.9) | 2468.8 (177.0) | 2623.5 (264.6) | 2372.5 (272.6) |
| | Cost | **2.1 (1.2)** | 3.6 (0.9) | 0.5 (0.9) | 99.7 (74.5) | 22.4 (5.3) |
| | Labeled Trajectories | **NA** | NA | 4349 (793) | 6000 (0) | 2749 (557) |
| Hopper | Reward | 1046.1 (345.2) | 1357.3 (403.5) | 1577.7 (42.1) | 1620.1 (139.9) | **1610.7 (48.9)** |
| | Cost | 29.1 (36.2) | **12.7 (7.6)** | 20.4 (44.2) | 284.2 (289.2) | **17.1 (4.7)** |
| | Labeled Trajectories | NA | NA | 6690 (736) | 6000 (0) | **2993 (484)** |
| Walker2d | Reward | 2682.2 (333.2) | 2562.8 (224.5) | **2797.7 (122.0)** | 2491.0 (478.3) | 2211.2 (182.0) |
| | Cost | **10.3 (5.5)** | **14.0 (4.8)** | **0.3 (0.3)** | 16.7 (9.1) | 24.1 (3.9) |
| | Labeled Trajectories | **NA** | **NA** | **15227 (1920)** | 6000 (0) | 5735 (1041) |

For a trajectory of length $T$, we score trajectories by the *conditional* CV of the trajectory-level surrogate cost under the estimator's sampling noise:

$$\mathrm{CV}_{\tau,w}\left[\sum_{t=0}^{T-1}\tilde{C}_t\right] := \frac{\sqrt{\mathrm{Var}\left[\sum_{t=0}^{T-1}\tilde{C}_t|\tau,w\right]}}{\mathbb{E}\left[\sum_{t=0}^{T-1}\tilde{C}_t|\tau,w\right]} = \frac{\sqrt{\sum_{t=0}^{T-1}\mathrm{Var}\left[\tilde{C}_t|\tau,w\right]}}{\sum_{t=0}^{T-1}\mathbb{E}\left[\tilde{C}_t|\tau,w\right]}.$$

The last equality holds because, conditional on $(\tau, w)$, the only remaining randomness is the independent per-step noise used to sample each $Z_t$; thus the sampled costs $\{\tilde{C}_t\}$ are conditionally independent and $\mathrm{Var}\left(\sum_t \tilde{C}_t \mid \tau, w\right) = \sum_t \mathrm{Var}(\tilde{C}_t \mid \tau, w)$ (Appendix A.2). We select trajectories with the largest $\mathrm{CV}_{\tau,w}$, targeting rollouts where the estimator's trajectory-level surrogate-cost estimate is unstable relative to its mean.

The **TraCeS** pseudo-code is provided in Appendix D.

# 5. Experimental Results

## 5.1. Experimental Setup

We evaluate **TraCeS** on 12 continuous-control tasks from three benchmark suites: MuJoCo (Todorov et al., 2012), Safety Gymnasium (Ji et al., 2023), and Bullet Safety Gym (Gronauer, 2022). Throughout, the agent does *not* observe the true cost or threshold at training time. Instead, we provide sparse binary accept/reject labels for trajectories or labeled prefixes, constructed from the oracle cost only

for benchmarking (experiment details in Appendix B.4).

**Baselines.** We compare against: (i) **PPO-Lagrangian** and **CPPO-PID** (Stooke et al., 2020) as oracle CMDP solvers that assume access to the true cost (reference ceilings); (ii) **RLSF** (Chirra et al., 2024), which learns a cost model from feedback but assumes the threshold is known; and (iii) a **Cost-Threshold baseline** (*C-T*), a simple MLP estimator for the fully zero-knowledge setting (Appendix B.2). We include **additional oracle CMDP solvers (CUP/FOCOPS/TRPO-Saute)** in Appendix C.1 since they assume access to the true cost and serve primarily as reference ceilings rather than comparable baselines in the zero-knowledge setting.

## 5.2. Main Results

We report results on a total of 8 tasks across Safety Gymnasium and MuJoCo in Table 1 (remaining 4 tasks in Appendix C.2), and group methods by what they assume about the constraint: *oracle* (cost and threshold known), *partial* (threshold known), and *zero-knowledge* (neither known).

Despite having *zero knowledge* of the cost function and threshold, **TraCeS** learns policies that satisfy the evaluation constraint (cost $\leq 25$) on most tasks while using substantially fewer labeled trajectories than the zero-knowledge *C-T* baseline. This supports the central claim that decomposing sparse trajectory labels into per-timestep violation

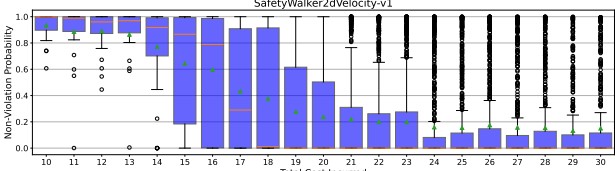

*Figure 3.* Walker2d: predicted trajectory-level non-violation probabilities $\hat{P}(\psi = 1 \mid \tau)$ versus oracle total cost (used only for benchmarking). Boxes show interquartile ranges across trajectories at each total-cost value; whiskers/outliers follow standard boxplot conventions; markers indicate the mean. Predictions drop rapidly as total cost crosses the benchmark threshold, with higher dispersion near the boundary.

credit improves feedback efficiency.

Relative to partial-knowledge RLSF, **TraCeS** achieves competitive reward–safety tradeoffs on most tasks, and remains stable in long-horizon settings where credit assignment is hardest. We also observe that oracle Lagrangian methods may violate the constraint on some tasks, consistent with known oscillatory behavior around the constraint boundary and prior benchmarking results (Ji et al., 2023). Results on Bullet Safety Gym (Appendix C.2) further indicate that **TraCeS** generalizes across dynamics and cost structures.

### 5.3. Qualitative Analysis

We qualitatively validate whether **TraCeS** learns (i) calibrated trajectory-level acceptability scores and (ii) localized per-timestep attribution, despite being trained only from sparse binary trajectory/prefix labels. We highlight Walker2d and include additional tasks in Appendix C.5.

**Trajectory-level acceptability estimates.** Figure 3 plots predicted non-violation probabilities $\hat{P}(\psi = 1 \mid \tau)$ against the oracle total cost (used only to construct labels for benchmarking). Predictions remain high for trajectories below the benchmark threshold and drop rapidly above it, with increased dispersion near the boundary. This indicates that the estimator's trajectory-level scores are consistent with the induced accept/reject labels.

**Timestep-level attribution.** Figure 4 compares the inferred per-step surrogate cost $\tilde{C}_t = -\log \hat{P}_t^{\Delta}$ (green; normalized for visualization) against the oracle cumulative cost (blue). Even though supervision is sparse and delayed, $\tilde{C}_t$ typically peaks near the violation event, defined as the first timestep $t^{\star}$ where oracle cumulative cost crosses the benchmark threshold. This suggests that the learned credits recover a localized notion of "what drove rejection" from coarse feedback.

**Attribution concentrates near violations and avoids zero-cost regions.** To quantify localization, we compute two

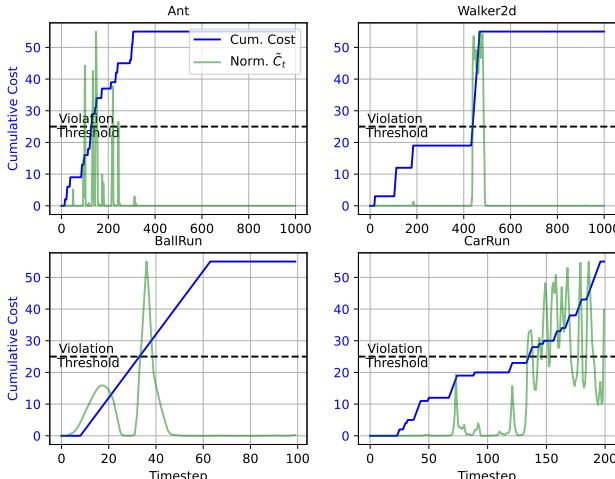

*Figure 4.* Per-timestep attribution examples: inferred surrogate cost $\tilde{C}_t = -\log \hat{P}_t^{\Delta}$ (green; normalized for visualization) versus oracle cumulative cost (blue) across domains (Ant, Walker2d, BallRun, CarRun). The dashed line is the benchmark violation threshold ($b = 25$). Even with only sparse binary labels for training, inferred costs tend to peak near the violation event (first threshold crossing), illustrating localized credit assignment.

complementary ratios. First, Figure 5 reports, for each trajectory, $\mathbb{E}[\tilde{C}_t \mid C_t = 0] / \mathbb{E}[\tilde{C}_t]$, where $C_t$ is the oracle per-step cost (benchmarking only). Ratios well below 1 indicate low attribution in true zero-cost timesteps. Second, Figure 6 reports the ratio between the mean $\tilde{C}_t$ in a 5-step window at offset $(t - t^{\star})$ and the trajectory-wide mean, $\mathbb{E}[\tilde{C}_t]$. Ratios $> 1$ near offset 0 show elevated attribution concentrated around the violation event. Together, these diagnostics indicate **TraCeS** learns localized credit assignment rather than spreading cost uniformly across the rollout.

**Additional tasks.** Appendix C.5 provides the same set of diagnostics for additional environments.

### 5.4. Additional Analysis

We provide additional ablations and robustness studies in the appendix: (i) low-data regime with a fixed labeling budget (Appendix C.3, C.6); (ii) random-initialization results without offline estimator pretraining (Appendix C.7); (iii) ablation of uncertainty-guided trajectory selection (Appendix C.8); (iv) sensitivity to $d$ and label noise (Appendix C.10, C.11); (v) sensitivity to numerical clamping, checkpoint sparsity, and terminal-only labels (Appendix C.12, C.13, C.14); and (vi) an alternative ground-truth threshold setting ($b = 50$; Appendix C.15).

**Human feedback.** To test robustness beyond simulator labels, we run a small-scale human annotation study on HalfCheetah, where a human labels whether the agent falls within 150 steps (Appendix C.9). With only 100 labeled

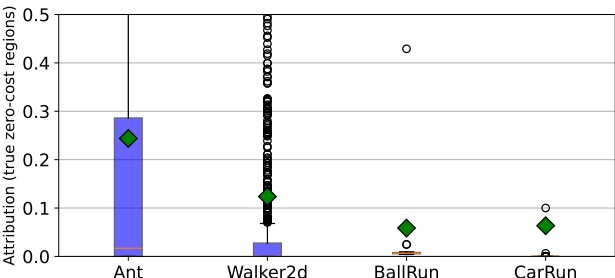

*Figure 5.* Attribution in zero-cost regions. For each trajectory, we compute $\frac{\mathbb{E}[\tilde{C}_t | C_t = 0]}{\mathbb{E}[\tilde{C}_t]}$, where $C_t$ is the oracle per-step cost used only for benchmarking. Ratios below 1 indicate low attribution in true zero-cost timesteps.

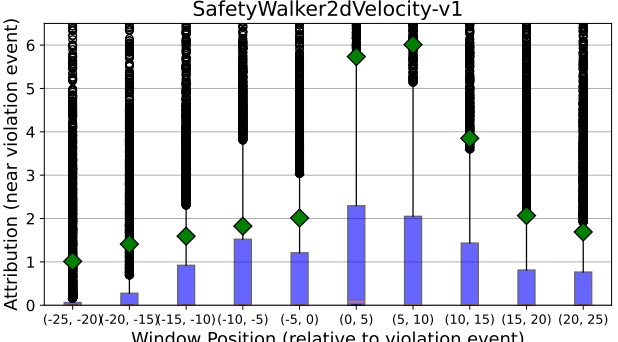

*Figure 6.* Attribution concentrates near the violation event. Let $t^\star$ be the first timestep where oracle cumulative cost crosses the benchmark threshold. We report the ratio $\frac{\text{mean } \tilde{C}_t \text{ in a 5-step window at offset } (t - t^\star)}{\text{mean } \tilde{C}_t \text{ over the trajectory}}$. Ratios $> 1$ around offset 0 indicate elevated attribution near the violation event.

trajectories per seed, **TraCeS** attains a higher non-violation rate than RLSF and the zero-knowledge baseline, suggesting the estimator remains effective under sparse, noisy human feedback. While limited in scale, this suggests per-timestep credits can be learned from coarse accept/reject judgments.

## 6. Conclusion

We presented **TraCeS**, a framework for learning constraint-satisfying behavior from sparse binary trajectory feedback when both the underlying cost function and its threshold are unknown. The key idea is to transform coarse accept/reject supervision into *per-timestep violation credit* via a sequential estimator whose multiplicative decomposition yields a trajectory-level non-violation probability. This credit signal induces a principled surrogate cost through a Jensen lower bound, enabling plug-in constrained policy optimization with standard CMDP solvers.

Empirically, **TraCeS** learns non-violating policies across multiple continuous-control benchmarks while using substantially fewer labeled trajectories than zero-knowledge baselines, and remains robust under low-data regimes, label noise, and sparser checkpoint feedback. Notably, our formulation naturally supports both terminal-only labels (a special case with a single checkpoint) and intermediate prefix labels, allowing practitioners to trade off annotation frequency against estimator sharpness.

This work suggests a practical path toward aligning RL agents with *implicit* and context-dependent constraints when formal specifications are unavailable. Promising directions include richer supervision beyond binary accept/reject (e.g., graded acceptability scores or ordinal severity levels), improved active labeling criteria beyond CV, and tighter integration with structured annotations that help localize violation time (e.g., short textual notes indicating what went wrong). We discuss limitations and failure modes in Appendix E.

## Acknowledgements

This research was supported by the National Research Foundation Singapore and DSO National Laboratories under the AI Singapore Programme (Award Number: AISG2RP-2020-016). We thank colleagues and the anonymous reviewers for helpful discussions and feedback.

## Impact Statement

This paper studies learning safety-aligned behavior in sequential decision-making from sparse trajectory-level feedback. The proposed method is domain-agnostic and could be applied in safety-critical settings such as robotics or autonomous systems, where improving adherence to implicit constraints may reduce unsafe behaviors. The experiments use standard simulation benchmarks and do not involve sensitive personal data. We do not anticipate immediate negative societal impacts beyond those generally associated with deploying RL systems in high-stakes environments.

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

# A. Theory

## A.1. Jensen's Slack Proof

**Theorem 7 (Jensen-Gap Bound).** Suppose $\hat{p}_\tau \in [\epsilon, 1]$ for all trajectories $\tau$, where $\hat{p}_\tau := \hat{P}(\psi = 1 \mid \tau)$, and $\epsilon > 0$ is ensured by clamping log-credits to be bounded below. Then,

$$\log\left(\mathbb{E}_{\tau \sim \pi_\theta}[\hat{p}_\tau]\right) - \mathbb{E}_{\tau \sim \pi_\theta}[\log \hat{p}_\tau] \leq \frac{1}{2\epsilon^2} \cdot \mathrm{Var}_{\tau \sim \pi_\theta}(\hat{p}_\tau)$$

*Proof of Theorem 7.* Let $f(x) = \log(x)$ and define $\mu := \mathbb{E}_{\tau \sim \pi_\theta}[\hat{p}_\tau]$. By Taylor's theorem with Lagrange remainder (Apostol, 1991; Rudin, 1976; Stewart, 2012), we expand $f(\hat{p}_\tau)$ around $\mu$:

$$f(\hat{p}_\tau) = f(\mu) + f'(\mu)(\hat{p}_\tau - \mu) + \frac{1}{2}f''(\xi)(\hat{p}_\tau - \mu)^2$$

for some intermediate $\xi$ between $\hat{p}_\tau$ and $\mu$, as required by the Lagrange form of the remainder. This expansion includes the Lagrange remainder term and is exact for some $\xi \in (\hat{p}_\tau, \mu)$. Taking expectations:

$$\mathbb{E}_{\tau \sim \pi_\theta}[f(\hat{p}_\tau)] = f(\mu) + f'(\mu)\mathbb{E}_{\tau \sim \pi_\theta}[\hat{p}_\tau - \mu] + \tfrac{1}{2}\mathbb{E}_{\tau \sim \pi_\theta}\left[f''(\xi)(\hat{p}_\tau - \mu)^2\right]$$

$$\mathbb{E}_{\tau \sim \pi_\theta}[f(\hat{p}_\tau)] = f(\mu) + \tfrac{1}{2}\mathbb{E}_{\tau \sim \pi_\theta}\left[f''(\xi)(\hat{p}_\tau - \mu)^2\right]$$

$$f(\mu) - \mathbb{E}_{\tau \sim \pi_\theta}[f(\hat{p}_\tau)] = \tfrac{1}{2}\mathbb{E}_{\tau \sim \pi_\theta}\left[-f''(\xi)(\hat{p}_\tau - \mu)^2\right]$$

The first-order term vanishes since $\mathbb{E}[\hat{p}_\tau - \mu] = 0$. Since $f(x) = \log(x)$, its second derivative satisfies $-f''(\xi) = \frac{1}{\xi^2} > 0$. Hence, the Jensen gap becomes:

$$\log(\mu) - \mathbb{E}_{\tau \sim \pi_\theta}[\log \hat{p}_\tau] = \frac{1}{2}\mathbb{E}_{\tau \sim \pi_\theta}\left[\frac{(\hat{p}_\tau - \mu)^2}{\xi^2}\right]$$

Since $\xi \in [\min\{\hat{p}_\tau, \mu\}, \max\{\hat{p}_\tau, \mu\}]$ and both $\hat{p}_\tau \geq \epsilon$, it follows that $\xi \geq \epsilon$, and hence $\frac{1}{\xi^2} \leq \frac{1}{\epsilon^2}$. So:

$$\log(\mathbb{E}_{\tau \sim \pi_\theta}[\hat{p}_\tau]) - \mathbb{E}_{\tau \sim \pi_\theta}[\log \hat{p}_\tau] \leq \frac{1}{2\epsilon^2} \cdot \mathbb{E}_{\tau \sim \pi_\theta}\left[(\hat{p}_\tau - \mu)^2\right] = \frac{1}{2\epsilon^2} \cdot \mathrm{Var}_{\tau \sim \pi_\theta}(\hat{p}_\tau)$$

$\square$

**Corollary 7.1 (Worst-Case Bound).** Applying Popoviciu's inequality with $\hat{p}_\tau \in [\epsilon, 1]$, we get:

$$\mathrm{Var}_{\tau \sim \pi_\theta}(\hat{p}_\tau) \leq \frac{(1-\epsilon)^2}{4} \quad \Rightarrow \quad \log(\mathbb{E}_{\tau \sim \pi_\theta}[\hat{p}_\tau]) - \mathbb{E}_{\tau \sim \pi_\theta}[\log \hat{p}_\tau] \leq \frac{(1-\epsilon)^2}{8\epsilon^2}$$

*Remark 8.* As training progresses, the estimator typically becomes more stable on the on-policy trajectory distribution, so $\mathrm{Var}(\hat{p}_\tau)$ decreases; consequently the Jensen gap shrinks.

## A.2. Conditional Variance Additivity for CV Scoring

In Section 4.3, the CV score is defined *conditional* on a fixed observed trajectory $\tau$ and fixed estimator parameters $w$. Given $(\tau, w)$, the estimator produces a deterministic sequence of distribution parameters $\{(\mu_t, \sigma_t)\}_{t=0}^{T-1}$ via the recurrent summary (e.g., $h_{t+1} = f_w(h_t, s_t, a_t)$ and $(\mu_t, \sigma_t) = g_w(h_t, h_{t+1})$). The sampled per-step log-credit can be written as

$$Z_t = -\exp(\mu_t + \sigma_t \epsilon_t), \qquad \epsilon_t \overset{iid}{\sim} \mathcal{N}(0, 1),$$

and we define $\tilde{C}_t := -Z_t \geq 0$. Conditional on $(\tau, w)$, the only randomness in $\tilde{C}_t$ comes from the independent noise draws $\{\epsilon_t\}$; hence the sampled costs $\{\tilde{C}_t\}_{t=0}^{T-1}$ are conditionally independent. Therefore,

$$\mathrm{Var}\left[\sum_{t=0}^{T-1} \tilde{C}_t \mid \tau, w\right] = \sum_{t=0}^{T-1} \mathrm{Var}\left[\tilde{C}_t \mid \tau, w\right].$$

We use this conditional variance (and the corresponding conditional mean) to form the trajectory-specific CV score in Eq. (Section 4.3). Note that *unconditional* independence need not hold because the parameters $(\mu_t, \sigma_t)$ vary with $\tau$ and with the evolving policy distribution.

# B. Additional Illustrations and Implementation/Experiment Details

## B.1. Visual Illustration of Challenges in Handcrafting Safety Cost

Figure 7 illustrates how trajectory-level safety constraint may contradict simple hand-designed cost function and threshold. Avoiding a pothole or wildlife animal may require driving off-lane temporarily. While these objects may be detectable with sensors, translating them into accurate, generalizable costs is nontrivial, as such costs depend on dynamic factors like pothole size, animal behavior, vehicle speed or even weather conditions. Handcrafted cost functions and safety thresholds (e.g., time spent off-lane or curvature penalties) often fail to generalize across dynamic and diverse safety scenarios. Moreoever, an over-engineered specification could potentially introduce unintended consequences.

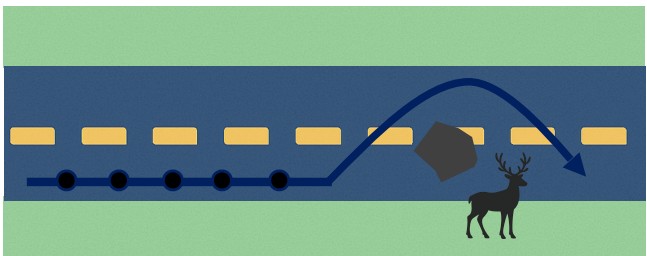

*Figure 7.* Acceptable behavior depends on dynamic, context-specific factors (e.g., pothole size, animal speed, or unexpected obstacles), making it difficult for handcrafted cost functions or static thresholds to generalize. This motivates learning violation/acceptability directly from trajectory-level feedback.

## B.2. (Zero Knowledge) Cost-Threshold (*C-T*) Baseline

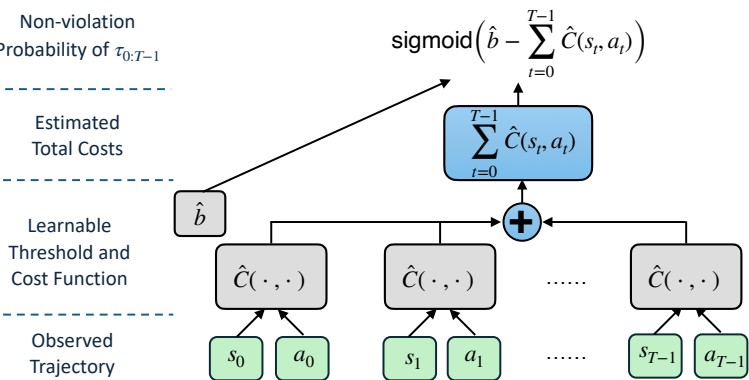

*Figure 8.* **C-T baseline violation estimator**. The same MLP estimates the non-negative cost $\hat{C}(\cdot, \cdot)$ at each timestep and $\hat{b}$ is the estimated threshold. A sigmoid function is then used to convert the remaining threshold into non-violation probability.

To the best of our knowledge, no prior work has addressed constraint learning with zero knowledge on cost and threshold. This motivates us to design a simple MLP baseline for our empirical evaluation. The model architecture of this baseline is depicted in Figure 8. This model uses a shared MLP to estimate the per-timestep cost directly and a threshold estimate $\hat{b}$ is also learned. The remaining threshold is then fed through a sigmoid function to compute the final non-violation probability for $\tau_{0:T-1}$.

As *C-T* baseline directly estimates the threshold and cost function, it solves the following CMDP program with constraint rewritten:

$$\max_{\theta} J_R(\pi_\theta) = \mathbb{E}_{\tau \sim \pi_\theta} \left[ \sum_{t=0}^{\infty} \gamma^t R(s_t, a_t) \right]$$

$$\text{s.t.} \quad J_c(\pi_\theta) = \mathbb{E}_{\tau \sim \pi_\theta} \left[ \sum_{t=0}^{\infty} \gamma^t \hat{C}(s_t, a_t) \right] \leq \hat{b} \tag{7}$$

Although *C-T* baseline provides a straightforward way to estimate costs and threshold, it can be tricky to have an accurate estimate as there could be many possible combinations of $\hat{b}$ and $\hat{C}(s, a)$ which minimize the binary cross entropy loss. Say a constant $k$ is added to $\hat{b}$, the estimated probability $\sigma(\hat{b} - \sum_{t=0}^{T_i-1} \hat{C}(s_t, a_t))$, will be unchanged if the same constant $k$ is added to $\sum_{t=0}^{T_i-1} \hat{C}(s_t, a_t)$. This could have contributed to its erratic performance in the main results.

## B.3. Implementation Details

### B.3.1. CONDITIONING THE CONSTRAINED COMPONENTS ON THE ESTIMATOR STATE

**TraCeS** augments a standard CMDP solver by providing a learned per-step surrogate cost $\tilde{C}_t = -\log \hat{P}_t^\Delta$, where $\hat{P}_t^\Delta$ is produced by the violation estimator. Because the estimator maintains a recurrent summary $h_t$ of the observed prefix, $\tilde{C}_t$ is, in general, a function of $(s_t, a_t, h_t)$ rather than $(s_t, a_t)$ alone. To expose this information to the CMDP solver, we condition the *constraint-related* components on $h_t$.

**Policy (actor).**   We parameterize the policy as $\pi_\theta(a_t \mid s_t, h_t)$, where $h_t$ is computed online by rolling the shared encoder forward along the trajectory (Fig. 2). Concretely, the policy network receives the concatenated input $[s_t; h_t]$ (and outputs an action distribution as usual). This allows the policy update to access the same prefix summary that the surrogate cost depends on.

**Constraint critic.**   Similarly, we parameterize the constraint value function as $V_{\tilde{C}}(s_t, h_t)$ (or $Q_{\tilde{C}}(s_t, a_t, h_t)$ depending on the solver), trained with targets constructed from the surrogate cost sequence $\{\tilde{C}_t\}$. This matches the solver's standard update rules, with the only change being that the critic input includes $h_t$.

**Reward critic.**   For the environments we benchmark (standard Markov control tasks), the reward is a function of $(s_t, a_t)$ and does not depend on the history beyond the current state. Accordingly, we keep the reward value function in its standard Markov form $V_R(s_t)$ (or $Q_R(s_t, a_t)$) without conditioning on $h_t$.[2]

**Summary.**   Overall, **TraCeS** does not change the underlying CMDP solver; it only (i) provides a learned per-step surrogate cost, and (ii) conditions the policy and constraint critic on the estimator's prefix summary $h_t$ to respect the dependence of $\tilde{C}_t$ on trajectory history.

## B.4. Other Experiment Details

### B.4.1. EXPERIMENT SETUP

The control tasks evaluated consist of 12 control tasks:

- **Four MuJoCo (Todorov et al., 2012) Tasks**: Agent's goal is to move to the right as quickly as possible while adhering to a velocity constraint. Each task has an episode horizon of 1000 timesteps.

- **Four Circle (Ji et al., 2023) Tasks**: Requires two agent types (Point and Car) to circle around a central area without leaving the boundaries for an extended period. Two difficulty levels are evaluated. Each task has an episode horizon of 500 timesteps.

- **Four Run (Gronauer, 2022) Tasks in Bullet Safety Gym**: Four agent types (Ant, Ball, Car, and Drone) must run through an avenue between safety boundaries. The BallRun task uses an episode horizon of 100 timesteps, while the AntRun, CarRun, and DroneRun tasks use 200 timesteps.

For all these 12 tasks, the ground-truth threshold is 25 and cost is a binary signal, consistent with the setup in Gronauer (2022); Ji et al. (2023). These information on the ground-truth constraint is not known to **TraCeS**. **TraCeS** only infer the constraint through training trajectories selected using **TraCeS** trajectory selection mechanism (Section 4.3): trajectories with highest CV (and above a threshold value) are selected for labeling. We highlight that **TraCeS** selects the trajectories

---

[2]Conditioning the reward critic on $h_t$ is optional and can be used as a variance-reduction feature; in our implementation it did not materially affect performance, so we keep the reward critic Markov for simplicity.

collected during RL training to retrain the estimator. In contrast, RLSF (Chirra et al., 2024) samples trajectories using another evaluation environment and subsequently selects them for cost learning.

Sparse binary labels are used to annotate these selected trajectories. For MuJoCo and Circle tasks, a prefix label is given every 20 timesteps. As for Run tasks, a segment label is given every 5 timesteps. Although long horizons make terminal-only credit assignment harder, this sparse checkpoint labeling provides intermediate localization signal while remaining much coarser than dense per-step supervision. A trajectory segment is only labeled non-violating if the true cost incurred so far is below the ground-truth threshold. The same sparse labeling strategy is used for our RLSF evaluation. Before commencing policy learning, both **TraCeS** estimator and RLSF cost module were pretrained with data from offline safe RL dataset (Liu et al., 2024).

To prevent catastrophic forgetting (Robins, 1995), both **TraCeS** and RLSF stores these labeled trajectory data into a buffer. To retrain **TraCeS** estimator (or RLSF cost learning module), we draw past labeled trajectory data from the buffer and retrain them together with new trajectory data. The retraining happens at interval during policy learning once sufficient number of new trajectories ($\geq 20$) are selected for annotation.

All experiments are run on a private GPU cloud, consisting of A5000, A40 and RTX3090 GPUs. Implementation of **TraCeS** policy learning is extended using omnisafe (Ji et al., 2024) codebase and RLSF evaluation is performed using code in Chirra et al. (2024).

**Computational overhead.** **TraCeS** adds two sources of computation relative to the underlying constrained policy optimizer: recurrent forward passes through the violation estimator during policy rollouts, and periodic estimator retraining when newly selected trajectories are labeled. In practice, this overhead is small because the estimator is lightweight (Table 2) and retraining is triggered only after enough new labeled trajectories have accumulated. At matched environment steps, **TraCeS** required 260.9s on CarRun compared to 249.9s for PPO-Lagrangian at 400K steps (+4.4%), and 1573.0s on Ant compared to 1512.4s at 1.0M steps (+4.0%). Most of the additional cost comes from periodic estimator retraining rather than per-step inference.

### B.4.2. PERFORMANCE REPORTING AND METRICS

The total number of policy learning steps is 10 million in all MuJoCo and Circle tasks, and 2 million for the Run tasks (except BallRun, which uses 1 million steps due to its shorter horizon). After policy learning completes, we evaluate the trained policy using 100 evaluation trajectories in separate environments. Reported metrics include total return, total true cost, and the number of binary-labeled full trajectories used to retrain the estimator. All table and chart results show the mean and standard deviation over 8 random seeds, unless otherwise stated.

*For completeness*, the initial pretraining accuracy of the **TraCeS** estimator (on held-out trajectory labels) ranges from 89% to 97% across tasks when trained offline, prior to RL policy optimization. Note that this value is only for reference, as the estimator is continually refined during policy learning and is not the main indicator of safety performance. In environments with noisy labels (Appendix C.11), initial estimator accuracy drops substantially due to label corruption; nevertheless, **TraCeS** remains robust to such noise, maintaining strong constraint satisfaction and reward performance.

B.4.3. HYPERPARAMETERS

The list of hyperparameters used in implementing **TraCeS** is summarized in Table 2. The policy learning part largely follows the hyperparameter settings used in Ji et al. (2024). The policy network outputs a Gaussian policy and its variance is part of the learnable parameters. The hyperparameter setting for RLSF can be found in Chirra et al. (2024).

| Hyperparameter | Value |
|---|---|
| Normalize Obs | True |
| Normalize Reward | True |
| Normalize Cost | True |
| Entropy Coefficient | 0.01 |
| Discount Rate $\gamma$ | 0.99 |
| GAE $\lambda$ Coefficient | 0.95 |
| PPO Clipping Ratio | 0.2 |
| Batch Size | 100 |
| Policy & Critic Network Size | [64, 64] MLP |
| Policy & Critic Activation Function | ReLU |
| Optimizer (Policy & Critic) | Adam |
| Policy & Critic Learning Rate | 0.0003 |
| Lagrange Multiplier Learning Rate | 0.035 |
| **TraCeS** Estimator Encoder | 2-Stack GRU with hidden dim = 4 |
| **TraCeS** Estimator Decoder | [64, 64] MLP |
| **TraCeS** Estimator Decoder Activation Function | ReLU |
| Optimizer (**TraCeS** Estimator) | Adam |
| **TraCeS** Estimator Learning Rate | 0.001 |

*Table 2.* Hyperparameter settings for **TraCeS** implementation.

## C. Additional Experimental Results

### C.1. Additional oracle CMDP solvers

We compare against additional constrained RL solvers that assume access to the oracle per-step cost and threshold: CUP (Yang et al., 2022), FOCOPS (Zhang et al., 2020), and TRPO-Saute (Sootla et al., 2022). These serve as reference ceilings and are not directly comparable to zero-knowledge methods.

*Table 3.* Safety gymnasium and MuJoCo performance (eval environment) – part 2. Non-violating policies are shown in black bold; the best among non-violating methods is highlighted in blue. Violating policies are shown in gray. Standard deviations shown in ().

| Task | Metric | CUP (Oracle) | FOCOPS (Oracle) | TRPO-Saute (Oracle) | TraCeS (Ours) (Zero Knowledge) |
|---|---|---|---|---|---|
| PointCircle1 | Reward | 44.3 (3.1) | 42.0 (5.9) | 39.6 (0.7) | 43.2 (1.5) |
| | Cost | 16.1 (12.9) | 16.3 (28.1) | 8.3 (7.8) | 16.2 (7.2) |
| | Labeled Trajectories | NA | NA | NA | 5830 (918) |
| PointCircle2 | Reward | 41.2 (0.4) | 39.0 (1.8) | 39.5 (1.0) | 41.1 (0.9) |
| | Cost | 31.8 (19.7) | 12.9 (9.3) | 3.7 (4.3) | 19.8 (9.5) |
| | Labeled Trajectories | NA | NA | NA | 5465 (1328) |
| CarCircle1 | Reward | 17.5 (0.7) | 18.6 (0.5) | 14.4 (0.5) | 17.4 (0.9) |
| | Cost | 27.5 (6.1) | 40.7 (12.8) | 26.4 (17.8) | 11.3 (8.0) |
| | Labeled Trajectories | NA | NA | NA | 4005 (7) |
| CarCircle2 | Reward | 15.4 (0.5) | 16.0 (0.6) | 12.8 (1.0) | 15.2 (0.9) |
| | Cost | 28.3 (10.7) | 29.2 (13.6) | 35.1 (23.0) | 11.9 (10.0) |
| | Labeled Trajectories | NA | NA | NA | 4341 (243) |
| Ant | Reward | 3119.7 (64.2) | 3177.6 (131.6) | 3102.0 (90.8) | 2885.4 (12.3) |
| | Cost | 5.1 (3.0) | 12.7 (5.5) | 17.0 (0.3) | 19.5 (5.3) |
| | Labeled Trajectories | NA | NA | NA | 4657 (551) |
| HalfCheetah | Reward | 2441.4 (663.4) | 2951.7 (92.5) | 2448.6 (498.9) | 2372.5 (272.6) |
| | Cost | 7.6 (10.1) | 3.1 (2.2) | 16.7 (1.0) | 22.4 (5.3) |
| | Labeled Trajectories | NA | NA | NA | 2749 (557) |
| Hopper | Reward | 1273.5 (541.2) | 1330.1 (421.0) | 1357.8 (296.2) | 1610.7 (48.9) |
| | Cost | 13.0 (7.4) | 31.9 (32.6) | 16.3 (4.3) | 17.1 (4.7) |
| | Labeled Trajectories | NA | NA | NA | 2993 (484) |
| Walker2d | Reward | 2388.1 (207.0) | 2455.3 (356.3) | 2138.7 (373.8) | 2211.2 (182.0) |
| | Cost | 6.6 (5.5) | 8.3 (5.1) | 18.1 (0.8) | 24.1 (3.9) |
| | Labeled Trajectories | NA | NA | NA | 5735 (1041) |

### C.2. Bullet Safety Gym results

We report results on the remaining 4 Bullet Safety Gym tasks not shown in Table 1. RLSF is omitted because the released implementation does not support this suite.

*Table 4.* Bullet safety gym performance (eval environment) – part 1. Non-violating policies are shown in black bold; the best among non-violating methods is highlighted in blue. Violating policies are shown in gray. Standard deviations shown in (). Note that RLSF implementation does not support bullet safety gym.

| Task | Metric | PPO-Lagrangian (Oracle) | CPPO-PID (Oracle) | *C-T* Baseline (Zero Knowledge) | TraCeS (Ours) (Zero Knowledge) |
|---|---|---|---|---|---|
| AntRun | Reward | 683.7 (24.7) | 675.5 (26.6) | 654.6 (39.8) | 590.5 (32.3) |
| | Cost | 25.4 (25.3) | 22.6 (12.2) | 33.3 (21.5) | 21.4 (1.9) |
| | Labeled Trajectories | NA | NA | 6000 (0) | 6521 (1008) |
| BallRun | Reward | 579.5 (332.2) | 455.2 (27.6) | 485.8 (29.7) | 457.0 (30.1) |
| | Cost | 17.8 (28.9) | 3.7 (6.3) | 35.3 (22.3) | 24.7 (0.8) |
| | Labeled Trajectories | NA | NA | 6000 (0) | 2627 (229) |
| CarRun | Reward | 577.7 (17.8) | 568.2 (2.7) | 569.1 (8.2) | 568.3 (9.4) |
| | Cost | 22.0 (8.9) | 14.8 (5.6) | 13.3 (11.9) | 23.3 (1.1) |
| | Labeled Trajectories | NA | NA | 6000 (0) | 8832 (988) |
| DroneRun | Reward | 459.0 (10.4) | 449.9 (4.5) | 446.4 (5.1) | 440.0 (3.3) |
| | Cost | 33.2 (8.1) | 10.2 (14.3) | 18.1 (7.2) | 13.6 (10.5) |
| | Labeled Trajectories | NA | NA | 6000 (0) | 3494 (541) |

*Table 5.* Bullet safety gym performance (eval environment) – part 2. Non-violating policies are shown in black bold; the best among non-violating methods is highlighted in blue. Violating policies are shown in gray. Standard deviations shown in (). Note that RLSF implementation does not support bullet safety gym.

| Task | Metric | CUP (Oracle) | FOCOPS (Oracle) | TRPO-Saute (Oracle) | TraCeS (Ours) (Zero Knowledge) |
|---|---|---|---|---|---|
| AntRun | Reward | 649.1 (40.1) | **684.8 (13.7)** | **652.0 (9.9)** | **590.5 (32.3)** |
| | Cost | 55.2 (47.6) | **23.3 (26.7)** | **13.8 (0.8)** | **21.4 (1.9)** |
| | Labeled Trajectories | NA | **NA** | NA | **6521 (1008)** |
| BallRun | Reward | 443.9 (35.0) | **603.2 (368.2)** | 135.8 (431.5) | **457.0 (30.1)** |
| | Cost | 27.0 (38.2) | **17.2 (28.9)** | 41.3 (34.3) | **24.7 (0.8)** |
| | Labeled Trajectories | NA | **NA** | NA | **2627 (229)** |
| CarRun | Reward | **567.8 (2.5)** | 573.7 (28.0) | **552.0 (59.8)** | **568.3 (9.4)** |
| | Cost | **6.9 (5.0)** | 44.9 (27.5) | **22.4 (20.4)** | **23.3 (1.1)** |
| | Labeled Trajectories | **NA** | NA | **NA** | **8832 (988)** |
| DroneRun | Reward | 381.0 (88.4) | 342.2 (168.5) | **447.1 (7.8)** | **440.0 (3.3)** |
| | Cost | 37.2 (47.7) | 53.0 (28.2) | **15.5 (22.5)** | **13.6 (10.5)** |
| | Labeled Trajectories | NA | NA | **NA** | **3494 (541)** |

Across additional tasks and benchmark suites, **TraCeS** shows a consistent pattern: using only sparse accept/reject labels, it learns a per-step surrogate cost that enables constrained policy optimization with substantially fewer labeled trajectories than the zero-knowledge *C-T* baseline, while achieving competitive reward–safety tradeoffs relative to partial-knowledge RLSF on many tasks (partial-knowledge method — RLSF samples fresh trajectories in a separate evaluation environment and selects them). The *C-T* baseline can be less stable: it relies on a per-step MLP cost model that does not explicitly represent history-dependent or survival-style structure, which can make attribution and downstream policy updates harder in long-horizon settings (Appendix B.2).

We emphasize that oracle constrained RL solvers (PPO-Lagrangian, CPPO-PID, CUP, FOCOPS, TRPO-Saute) assume access to the true per-step cost and threshold; we report them as reference ceilings rather than as directly comparable baselines in the zero-knowledge setting.

## C.3. Matching Label Budget Used by RLSF

**TraCeS** selects training trajectories with CV higher than certain threshold for labeling and RLSF uses its novelty estimation method to select trajectories to annotate. In both scenarios, the setting does not correspond to a fixed number of labeled trajectories and thus the number of labeled trajectories can vary in Table 1 and Table 4. To ensure that **TraCeS** is compared with RLSF fairly with similar number of labeled trajectories, we fixed the maximum number of training trajectories selected by **TraCeS** using trajectory CV scores. We retest PointCircle1, PointCircle2 and Walker2d tasks and tabulate the result in Table 6. Result in Table 6 shows that **TraCeS** produces non-violating policies with reward matching that of RLSF which uses slightly more labeled trajectories, demonstrating **TraCeS** label efficiency.

*Table 6.* Performance (eval environment) with number of labeled trajectories provided to **TraCeS** fixed to slightly fewer than RLSF. Non-violating policies are shown in black bold; the best among non-violating methods is highlighted in blue. Violating policies are shown in gray. Standard deviations shown in ().

| Task | Metric | PPO-Lagrangian (Oracle) | RLSF (Partial Knowledge) | *C-T* Baseline (Zero Knowledge) | TraCeS (Ours) (Zero Knowledge) |
|---|---|---|---|---|---|
| PointCircle1 | Reward | **46.6 (1.4)** | **41.9 (1.8)** | 40.2 (1.1) | **40.6 (2.8)** |
| | Cost | **21.5 (7.2)** | **1.6 (2.2)** | 25.3 (15.9) | **15.5 (8.9)** |
| | Labeled Trajectories | **NA** | 3244 (966) | 12000 (0) | **3000 (0)** |
| PointCircle2 | Reward | 41.7 (0.9) | **39.9 (0.8)** | **40.2 (0.8)** | **39.9 (1.8)** |
| | Cost | 29.1 (6.00) | **2.6 (4.5)** | **22.7 (13.5)** | **22.2 (16.6)** |
| | Labeled Trajectories | NA | 3422 (589) | **12000 (0)** | **3000 (0)** |
| Walker2d | Reward | **2682.2 (333.2)** | **2797.7 (122.0)** | 2491.0 (478.3) | **2329.4 (309.1)** |
| | Cost | **10.3 (5.5)** | **0.3 (0.3)** | 16.7 (9.1) | **20.7 (2.0)** |
| | Labeled Trajectories | **NA** | **15227 (1920)** | **6000 (0)** | **10000 (0)** |

## C.4. Training Curves

We provide the training curves for PPO-Lagrangian, *C-T* baseline and **TraCeS** in Figure 9, 10, and 11. The solid line represents the mean value while the semi-transparent band indicates one standard-deviation across seeds. Note that these diagrams plot the reward and cost attained in the **training environment**, not the evaluation performance reported in Table 1 and 4.

In all cases, we observe from the figures that **TraCeS** reliably converges to non-violating policy (cost below the black dashed line, i.e., true threshold 25) as training progresses. Unlike the oracle method, it has to learn the constraint and hence it progressively converges to a non-violating and quality policy as training advances. In contrast, the *C-T* baseline fails to consistently produce non-violating policy in most tasks. Even in the tasks when *C-T* baseline manages to learn a non-violating policy, it takes much longer to converge than **TraCeS**.

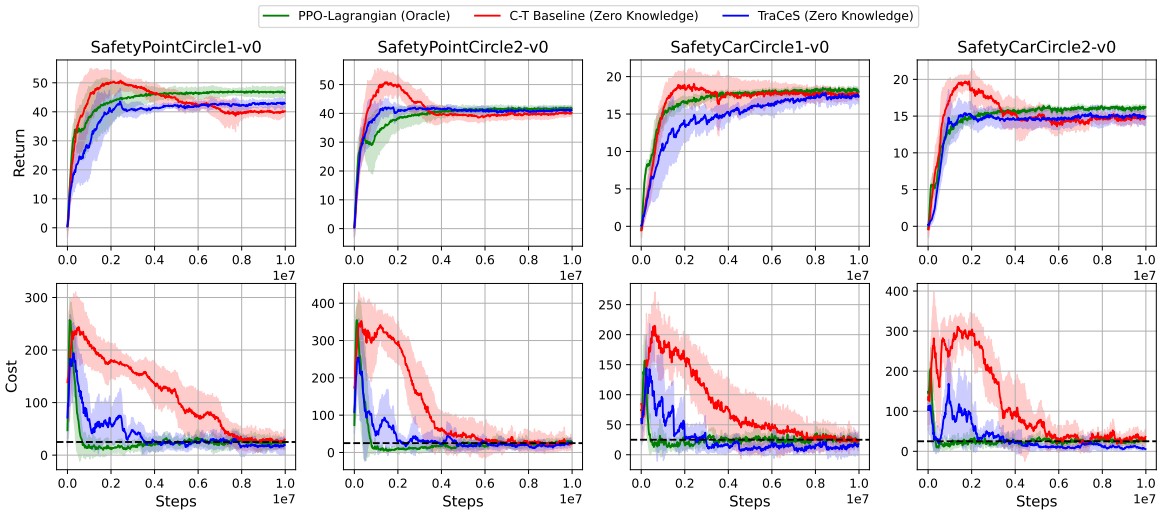

*Figure 9.* **Circle task training curve.** Solid line shows the mean; shaded region indicates one standard deviation across seeds.

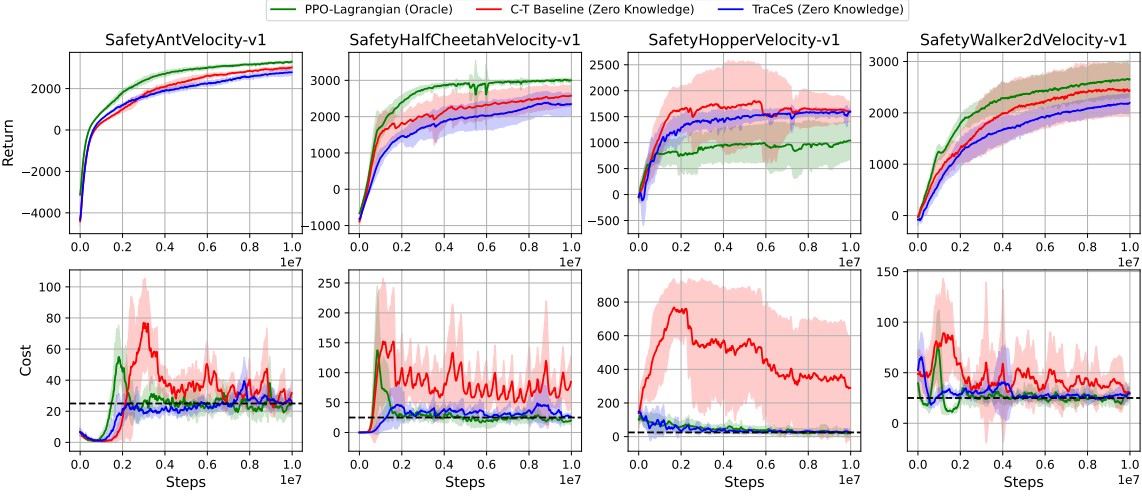

*Figure 10.* **MuJoCo task training curve.** Solid line shows the mean; shaded region indicates one standard deviation across seeds.

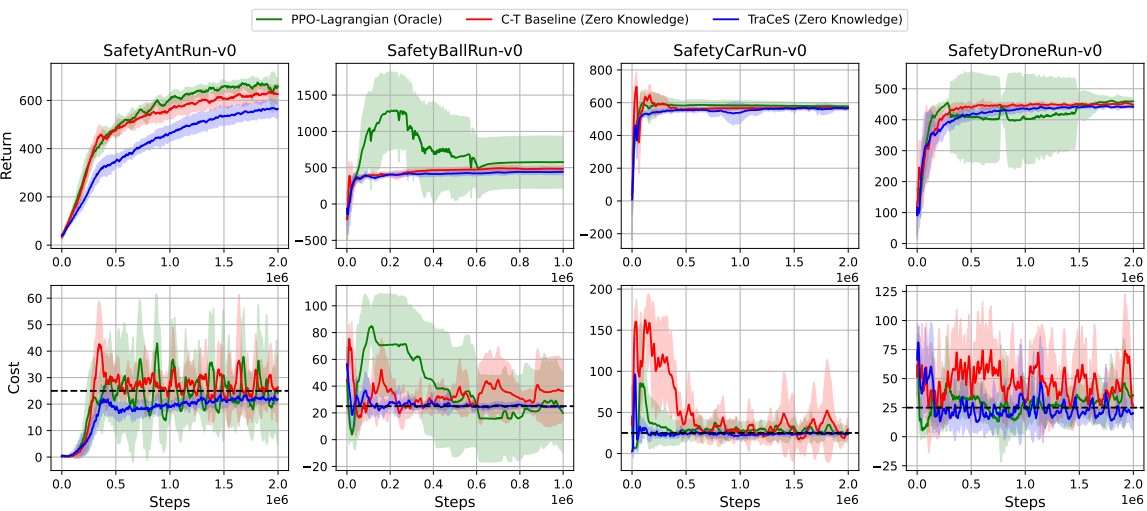

*Figure 11.* **Run task training curve.** Solid line shows the mean; shaded region indicates one standard deviation across seeds.

## C.5. Additional Qualitative Analysis

We provide additional qualitative analysis (violation credit assignment) result here. Figure 12, 13, 14, and 15 depict the distribution of trajectory-level non-violation probability estimates in Walker2d and three additional tasks. We observe similar pattern as the main paper (Section 5: **Qualitative Analysis**), where trajectories with total cost below the true threshold ($\leq 25$) are assigned higher non-violation probability by the estimator. At the same time, **TraCeS** estimator predicts low non-violation probability for the trajectories with total cost higher than true threshold. This agrees with the analysis in the main paper.

Detailed timestep-level attribution for 3 additional tasks is plotted in Figure 16, 17, and 18. From these diagrams, we can draw similar conclusion as the main paper since the attribution around the violation event (when total incurred cost crosses 25) is mostly concentrated well above 1. **TraCeS** estimator captures the timestep critical to violation and infers high surrogate cost around this region, suggesting accurate temporal violation credit assignment.

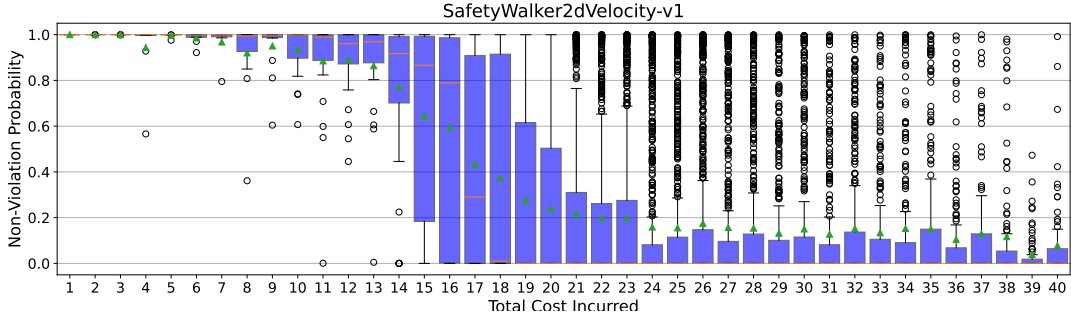

*Figure 12.* Walker2d: predicted trajectory-level non-violation probabilities $\hat{P}(\psi = 1 \mid \tau)$ versus oracle total cost (used only for benchmarking). Boxes show interquartile ranges across trajectories at each total-cost value; whiskers/outliers follow standard boxplot conventions; markers indicate the mean. Predictions drop rapidly as total cost crosses the benchmark threshold, with higher dispersion near the boundary.

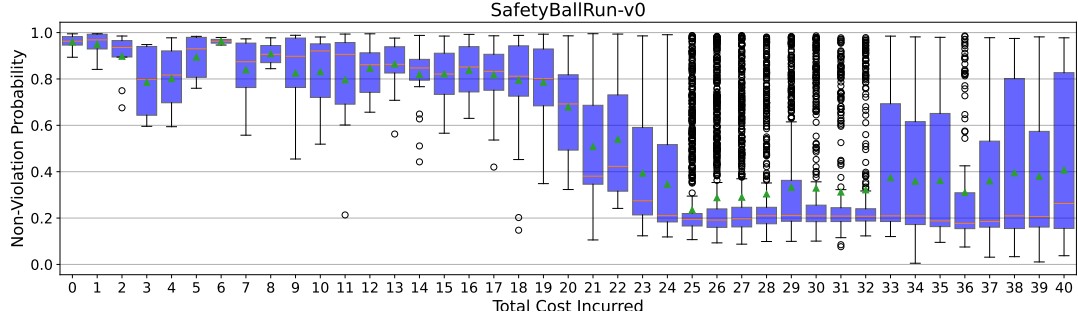

*Figure 13.* BallRun: predicted trajectory-level non-violation probabilities $\hat{P}(\psi = 1 \mid \tau)$ versus oracle total cost (used only for benchmarking). Boxes show interquartile ranges across trajectories at each total-cost value; whiskers/outliers follow standard boxplot conventions; markers indicate the mean. Predictions drop rapidly as total cost crosses the benchmark threshold, with higher dispersion near the boundary.

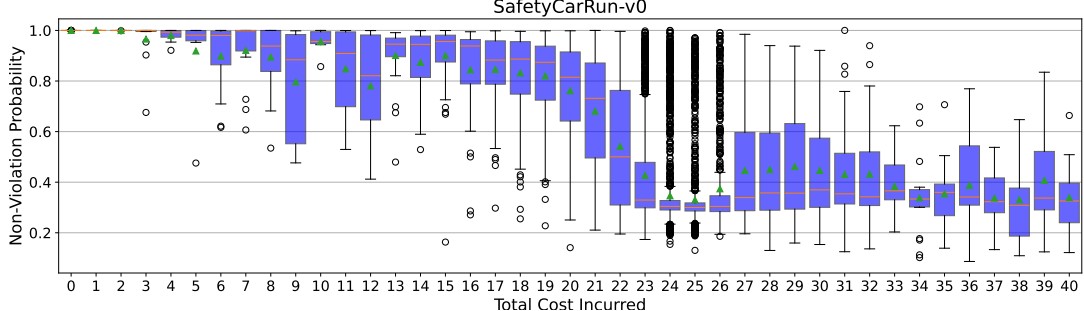

*Figure 14.* CarRun: predicted trajectory-level non-violation probabilities $\hat{P}(\psi = 1 \mid \tau)$ versus oracle total cost (used only for benchmarking). Boxes show interquartile ranges across trajectories at each total-cost value; whiskers/outliers follow standard boxplot conventions; markers indicate the mean. Predictions drop rapidly as total cost crosses the benchmark threshold, with higher dispersion near the boundary.

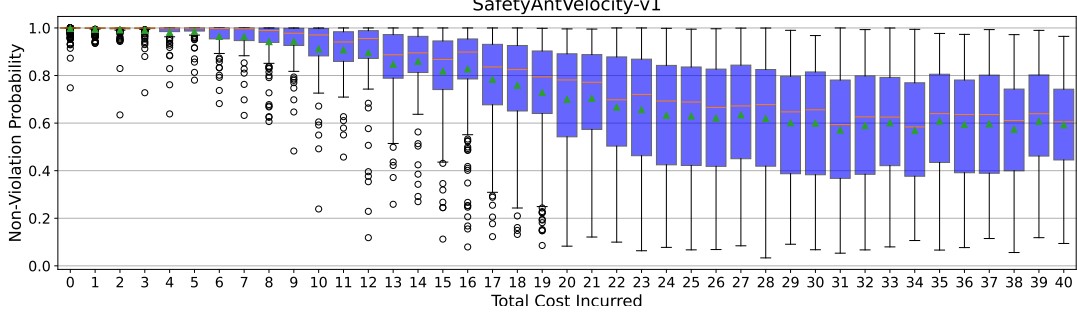

*Figure 15.* Ant: predicted trajectory-level non-violation probabilities $\hat{P}(\psi = 1 \mid \tau)$ versus oracle total cost (used only for benchmarking). Boxes show interquartile ranges across trajectories at each total-cost value; whiskers/outliers follow standard boxplot conventions; markers indicate the mean. Predictions drop rapidly as total cost crosses the benchmark threshold, with higher dispersion near the boundary.

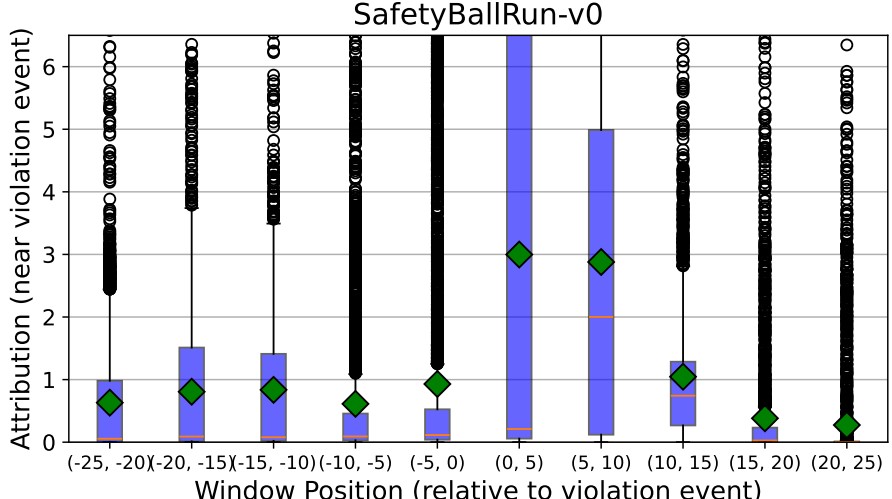

*Figure 16.* BallRun: Attribution concentrates near the violation event. Let $t^\star$ be the first timestep where oracle cumulative cost crosses the benchmark threshold. We report the ratio $\frac{\text{mean } \tilde{C}_t \text{ in a 5-step window at offset } (t-t^\star)}{\text{mean } \tilde{C}_t \text{ over the trajectory}}$. Ratios $> 1$ around offset 0 indicate elevated attribution near the violation event.

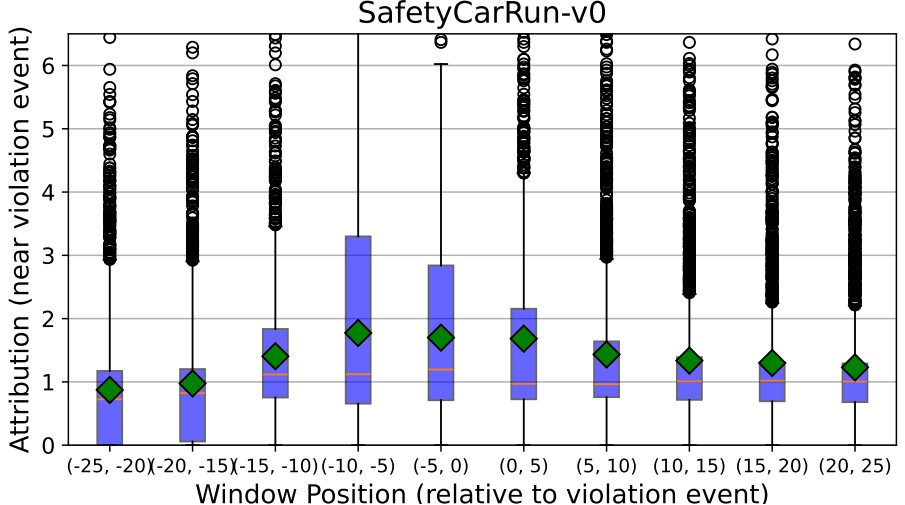

*Figure 17.* CarRun: Attribution concentrates near the violation event. Let $t^\star$ be the first timestep where oracle cumulative cost crosses the benchmark threshold. We report the ratio $\frac{\text{mean } \tilde{C}_t \text{ in a 5-step window at offset } (t-t^\star)}{\text{mean } \tilde{C}_t \text{ over the trajectory}}$. Ratios $> 1$ around offset 0 indicate elevated attribution near the violation event.

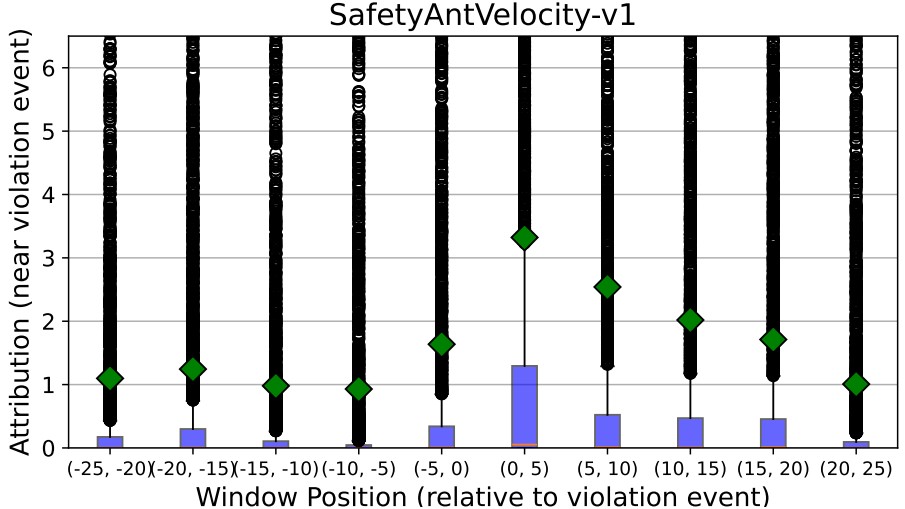

*Figure 18.* Ant: Attribution concentrates near the violation event. Let $t^\star$ be the first timestep where oracle cumulative cost crosses the benchmark threshold. We report the ratio $\frac{\text{mean } \tilde{C}_t \text{ in a 5-step window at offset } (t-t^\star)}{\text{mean } \tilde{C}_t \text{ over the trajectory}}$. Ratios $> 1$ around offset 0 indicate elevated attribution near the violation event.

## C.6. Low Data Regime

We evaluate **TraCeS** in a low-data regime with a fixed labeling budget of 1000 labeled trajectories per run. We select up to 1000 trajectories for labeling using the CV criterion (Section 4.3); *C-T* is also restricted to 1000 labeled trajectories. Tables 7 and 8 report evaluation performance.

RLSF does not expose a straightforward knob to cap the number of labeled trajectories in our implementation, so we report its default results for reference. Under this strict labeling budget, **TraCeS** typically remains constraint-satisfying and retains competitive reward, whereas the zero-knowledge *C-T* baseline is substantially less reliable across tasks.

*Table 7.* Low data regime performance in safety gymnasium and MuJoCo (eval environment). The labeling budget is capped at 1000 labeled trajectories for both **TraCeS** and *C-T* baseline. Non-violating policies are shown in black bold; the best among non-violating methods is highlighted in blue. Violating policies are shown in gray. Standard deviations shown in ().

| Task | Metric | PPO-Lagrangian (Oracle) | RLSF (Partial Knowledge) | *C-T* Baseline (Zero Knowledge) | **TraCeS (Ours)** (Zero Knowledge) |
|---|---|---|---|---|---|
| PointCircle1 | Reward | **46.6 (1.4)** | **41.9 (1.8)** | 40.8 (9.0) | **41.4 (1.4)** |
| | Cost | **21.5 (7.2)** | **1.6 (2.2)** | 82.2 (30.6) | **23.6 (12.1)** |
| | Labeled Trajectories | **NA** | 3244 (966) | 1000 (0) | **1000 (0)** |
| PointCircle2 | Reward | 41.7 (0.9) | **39.9 (0.8)** | 39.4 (1.6) | **39.5 (2.5)** |
| | Cost | 29.1 (6.00) | **2.6 (4.5)** | 39.3 (34.3) | **22.1 (17.7)** |
| | Labeled Trajectories | NA | **3422 (589)** | 1000 (0) | **1000 (0)** |
| CarCircle1 | Reward | **18.3 (0.4)** | 14.3 (2.2) | 15.8 (1.6) | **15.8 (1.0)** |
| | Cost | **23.5 (5.9)** | 33.8 (41.7) | 29.8 (9.7) | **13.3 (12.5)** |
| | Labeled Trajectories | **NA** | 7479 (645) | 1000 (0) | **1000 (0)** |
| CarCircle2 | Reward | 16.2 (0.2) | **13.7 (1.1)** | 14.7 (1.1) | **15.1 (1.1)** |
| | Cost | 28.3 (5.2) | **20.2 (17.0)** | 77.9 (34.1) | **22.6 (17.6)** |
| | Labeled Trajectories | NA | 7601 (597) | 1000 (0) | **1000 (0)** |
| Ant | Reward | **3313.1 (53.8)** | 2220.7 (91.8) | 3073.7 (52.8) | **2522.1 (150.1)** |
| | Cost | **14.4 (6.8)** | **5.8 (2.6)** | 39.0 (9.7) | **16.3 (4.7)** |
| | Labeled Trajectories | **NA** | 9036 (1289) | 1000 (0) | **1000 (0)** |
| HalfCheetah | Reward | **3008.3 (52.9)** | 2468.8 (177.0) | 2567.2 (138.3) | **2085.9 (422.9)** |
| | Cost | **2.1 (1.2)** | **0.5 (0.9)** | 102.2 (62.8) | **21.3 (2.6)** |
| | Labeled Trajectories | **NA** | 4349 (793) | 1000 (0) | **1000 (0)** |
| Hopper | Reward | 1046.1 (345.2) | **1577.7 (42.1)** | 1782.8 (487.9) | **1426.3 (122.7)** |
| | Cost | 29.1 (36.2) | **20.4 (44.2)** | 486.7 (410.1) | **18.0 (4.7)** |
| | Labeled Trajectories | NA | **6690 (736)** | 1000 (0) | **1000 (0)** |
| Walker2d | Reward | **2682.2 (333.2)** | **2797.7 (122.0)** | 2657.3 (405.6) | **1656.9 (227.0)** |
| | Cost | **10.3 (5.5)** | **0.3 (0.3)** | 32.4 (8.0) | **22.6 (8.1)** |
| | Labeled Trajectories | **NA** | **15227 (1920)** | 1000 (0) | **1000 (0)** |

*Table 8.* Low data regime performance in bullet safety gym (eval environment). The number of labeled trajectories is fixed to 1000 for both **TraCeS** and *C-T* baseline. Non-violating policies are shown in black bold; the best among non-violating methods is highlighted in blue. Violating policies are shown in gray. Standard deviations shown in (). Note that RLSF implementation does not support bullet safety gym.

| Task | Metric | PPO-Lagrangian (Oracle) | *C-T* Baseline (Zero Knowledge) | **TraCeS (Ours)** (Zero Knowledge) |
|---|---|---|---|---|
| AntRun | Reward | 683.7 (24.7) | 635.5 (24.5) | **494.0 (63.4)** |
| | Cost | 25.4 (25.3) | 34.2 (11.3) | **23.0 (4.0)** |
| | Labeled Trajectories | NA | 1000 (0) | **1000 (0)** |
| BallRun | Reward | **579.5 (332.2)** | **440.7 (24.3)** | **413.8 (35.2)** |
| | Cost | **17.8 (28.9)** | **22.5 (7.8)** | **24.3 (2.8)** |
| | Labeled Trajectories | **NA** | **1000 (0)** | **1000 (0)** |
| CarRun | Reward | **577.7 (17.8)** | **566.7 (4.2)** | **566.4 (5.8)** |
| | Cost | **22.0 (8.9)** | **8.0 (8.6)** | **21.4 (4.8)** |
| | Labeled Trajectories | **NA** | **1000 (0)** | **1000 (0)** |
| DroneRun | Reward | 459.0 (10.4) | 444.7 (4.3) | **422.6 (12.4)** |
| | Cost | 33.2 (8.1) | 45.8 (28.1) | **7.8 (8.1)** |
| | Labeled Trajectories | NA | 1000 (0) | **1000 (0)** |

## C.7. Random Initialization Without Offline Estimator Pretraining

In the main experiments, we warm-start the violation estimator using an offline set of labeled trajectories before entering the iterative estimator–policy loop. This improves sample efficiency and stabilizes early policy updates. To test whether **TraCeS** fundamentally depends on a high-quality pretrained estimator, we run an additional ablation that starts policy optimization with a randomly initialized violation estimator, while keeping the subsequent iterative training procedure unchanged.

Table 9 shows that **TraCeS** still learns viable non-violating policies on representative tasks from MuJoCo and Bullet Safety Gym. Performance is weaker than the corresponding pretrained setting, as expected, because the early surrogate costs are noisier and provide less reliable guidance to the policy optimizer. Nevertheless, the policies remain below the benchmark violation threshold across all four tasks, suggesting that offline estimator pretraining mainly improves sample efficiency and stability rather than being required for the iterative estimator–policy loop to succeed.

*Table 9.* Performance of **TraCeS** when starting from a randomly initialized violation estimator, without offline estimator pretraining. Results are reported in the evaluation environment. Non-violating policies are shown in black bold. Standard deviations are shown in parentheses.

| Task | Reward | Cost |
|------|--------|------|
| Ant | **2655.3 (201.0)** | **17.7 (2.7)** |
| HalfCheetah | **1934.5 (439.2)** | **22.9 (2.7)** |
| AntRun | **500.3 (35.9)** | **23.5 (1.9)** |
| CarRun | **559.0 (17.1)** | **22.3 (3.4)** |

## C.8. Ablation: Trajectory Selection

To assess the impact of our CV-based selective labeling strategy (Section 4.3), we run a modified version of **TraCeS** , which randomly selects a subset of collected trajectories for labeling, and compare against **TraCeS** with CV selection strategy. The total number of labeled trajectories is fixed to 1000 in both cases and results are reported in Table 10.

CV-based selection yields non-violating or near-threshold policies across the full suite and often improves reward, whereas random selection fails to satisfy the constraint on Hopper and BallRun. These tasks are particularly challenging under sparse feedback because early rollouts can contain large, concentrated violations, making it important to prioritize informative trajectories for estimator updates. The CV criterion surfaces trajectories where the estimator is most uncertain, improving data efficiency when labels are scarce.

These results further demonstrate the value of our uncertainty-guided trajectory selection strategy for learning non-violating policies with few labeled trajectories (low data regime). To further illustrate this, we plot their training curves in Figure 19: CV selection is more stable, achieves lower variance, and converges to non-violating behavior faster. In contrast, random selection often results in oscillatory training or delayed constraint satisfaction.

In the remaining tasks where both variants satisfy the constraint, CV-based selection often achieves slightly higher reward. When randomly sampled trajectories are already sufficiently informative for training the estimator, the marginal benefit of selection is smaller; nonetheless, prioritizing high-uncertainty rollouts can still improve stability and label efficiency.

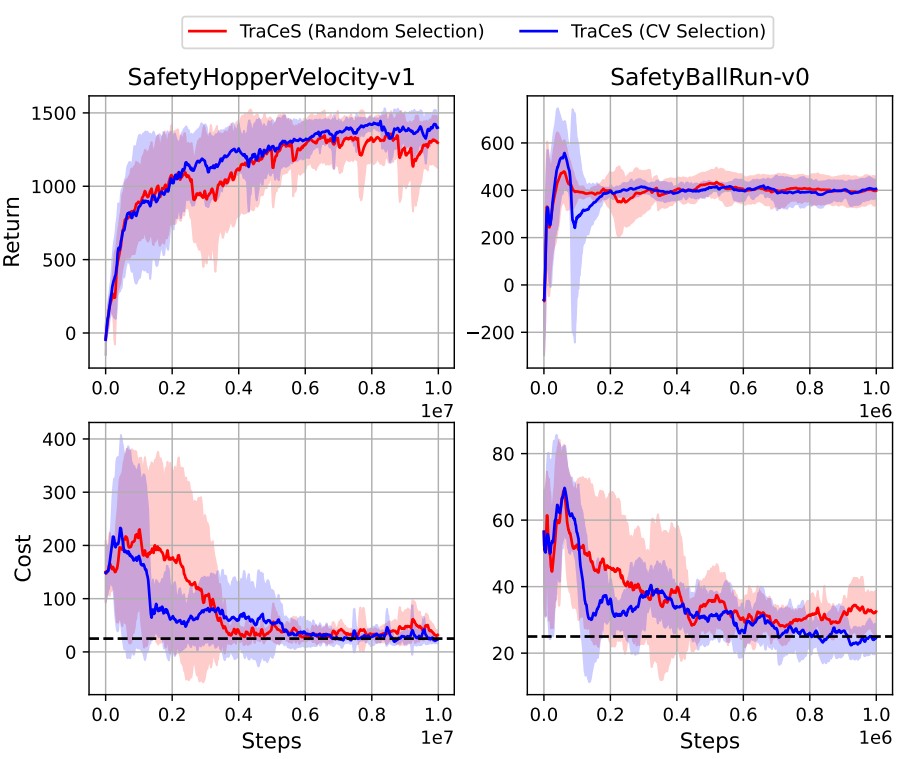

*Figure 19.* **Hopper and BallRun training curves (ablation study on trajectory selection).** Solid line shows the mean; shaded region indicates one standard deviation across seeds. CV selection strategy is more stable and converges to a non-violating policy sooner than random selection.

*Table 10.* Ablation study on trajectory selection (eval environment). The number of labeled trajectories is fixed to 1000 for both **TraCeS** (random selection) and **TraCeS** (CV selection). Non-violating policies are shown in black bold; the best among non-violating methods is highlighted in blue. Violating policies are shown in gray. Standard deviations shown in ().

| Task | Metric | **TraCeS** (Random Selection) | **TraCeS** (CV Selection) |
|---|---|---|---|
| PointCircle1 | Reward | **40.1 (3.3)** | **41.4 (1.4)** |
| | Cost | **17.0 (8.9)** | **23.6 (12.1)** |
| | Labeled Trajectories | **1000 (0)** | **1000 (0)** |
| PointCircle2 | Reward | **40.8 (1.0)** | **39.5 (2.5)** |
| | Cost | **17.0 (9.3)** | **22.1 (17.7)** |
| | Labeled Trajectories | **1000 (0)** | **1000 (0)** |
| CarCircle1 | Reward | **15.3 (1.1)** | **15.8 (1.0)** |
| | Cost | **15.3 (7.7)** | **13.3 (12.5)** |
| | Labeled Trajectories | **1000 (0)** | **1000 (0)** |
| CarCircle2 | Reward | **14.3 (1.1)** | **15.1 (1.1)** |
| | Cost | **14.8 (7.2)** | **22.6 (17.6)** |
| | Labeled Trajectories | **1000 (0)** | **1000 (0)** |
| Ant | Reward | **2497.8 (137.4)** | **2522.1 (150.1)** |
| | Cost | **20.1 (4.5)** | **16.3 (4.7)** |
| | Labeled Trajectories | **1000 (0)** | **1000 (0)** |
| HalfCheetah | Reward | **2151.6 (301.1)** | **2085.9 (422.9)** |
| | Cost | **22.4 (5.4)** | **21.3 (2.6)** |
| | Labeled Trajectories | **1000 (0)** | **1000 (0)** |
| Hopper | Reward | 1292.6 (209.5) | **1426.3 (122.7)** |
| | Cost | 31.1 (22.7) | **18.0 (4.7)** |
| | Labeled Trajectories | 1000 (0) | **1000 (0)** |
| Walker2d | Reward | **1361.3 (385.5)** | **1656.9 (227.0)** |
| | Cost | **18.9 (5.9)** | **22.6 (8.1)** |
| | Labeled Trajectories | **1000 (0)** | **1000 (0)** |
| AntRun | Reward | **459.0 (39.0)** | **494.0 (63.4)** |
| | Cost | **24.6 (4.1)** | **23.0 (4.0)** |
| | Labeled Trajectories | **1000 (0)** | **1000 (0)** |
| BallRun | Reward | 401.9 (56.0) | **413.8 (35.2)** |
| | Cost | 30.8 (6.1) | **24.3 (2.8)** |
| | Labeled Trajectories | 1000 (0) | **1000 (0)** |
| CarRun | Reward | **558.9 (6.3)** | **566.4 (5.8)** |
| | Cost | **20.0 (4.0)** | **21.4 (4.8)** |
| | Labeled Trajectories | **1000 (0)** | **1000 (0)** |
| DroneRun | Reward | **410.5 (21.6)** | **422.6 (12.4)** |
| | Cost | **10.2 (7.7)** | **7.8 (8.1)** |
| | Labeled Trajectories | **1000 (0)** | **1000 (0)** |

## C.9. Human Annotation Experiment

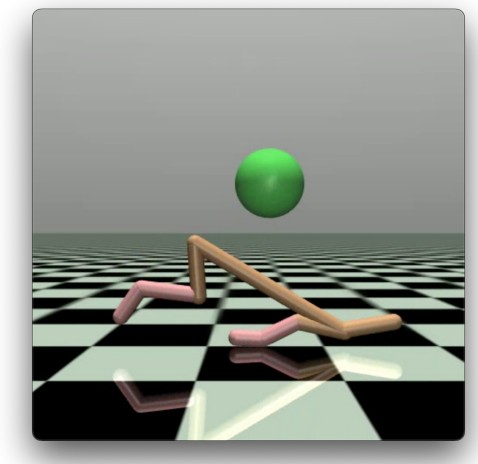

*(a)* HalfCheetah falling forward

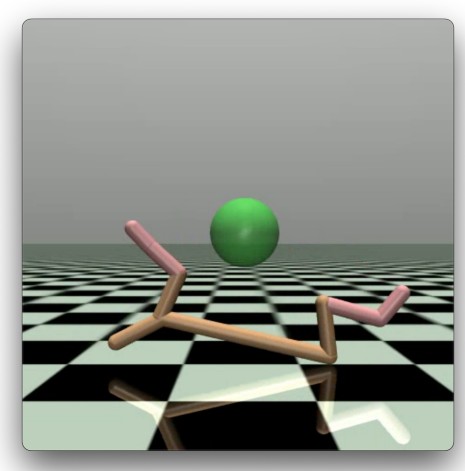

*(b)* HalfCheetah falling backward

*Figure 20.* HalfCheetah may fall forward or backward if it loses balance. Such falls indicate constraint-violating behavior for the purpose of human annotation.

We evaluate **TraCeS**'s robustness to real-world supervision by conducting a human feedback experiment on HalfCheetah. To isolate the effect of trajectory-level annotation, reward learning is disabled to ensure policy learning is driven solely by feedback. We treat "staying upright" as the primary objective in this experiment; the policy is optimized using the learned safety signal rather than environment reward. We compare **TraCeS** to RLSF and *C-T* baseline.

The agent's goal is to remain balanced for 150 timesteps without falling forward or backward (examples in Figure 20). A human annotator labels a trajectory as constraint-violated if a fall is observed within the 150 timesteps; otherwise, it is labeled as non-violated. Our annotation protocol closely follows prior works (Hejna & Sadigh, 2023; Hejna et al., 2024; Kim et al., 2023), where each rollout is visualized and labeled without revealing the policy source. This labeling protocol ensures a fair evaluation process. During evaluation, the human annotator remains unaware of which method generated each trajectory. This helps prevent bias.

We collect 100 labeled trajectories per seed and evaluate over 3 random seeds. For each method, the annotator reviews 10 evaluation rollouts per seed and reports the percentage of non-violating trajectories (i.e., no falls within 150 steps). The results (mean and standard deviation) are shown in Table 11. **TraCeS** outperforms both baselines, achieving a higher percentage of non-violating trajectories, demonstrating greater robustness to sparse and noisy human feedback.

*Table 11.* Human experiment (eval environment). All methods are trained using 100 labeled trajectories. The best-performing method is shown in blue; others are in black. Standard deviations shown in ().

|  | RLSF | *C-T* Baseline | **TraCeS (Ours)** |
|---|---|---|---|
| Non-Violating Trajectory % | 50.0 (8.2) | 33.3 (26.2) | **66.7 (17.0)** |

## C.10. Sensitivity Analysis of Hyperparameter $d$

We investigate the sensitivity of **TraCeS** to the desired acceptability rate hyperparameter $d$, which specifies the minimum proportion of non-violating trajectories required for policy feasibility (see Section 4.2 for formulation). To assess robustness, we vary $d$ from a very strict value of 0.99 to a relaxed threshold of 0.5, while fixing the number of labeled trajectories to 1000 for all runs.

Table 12 summarizes the results for four representative tasks. Across $d \in \{0.99, 0.9, 0.8\}$, **TraCeS** shows similar perfor-

mance, suggesting limited sensitivity in a practically relevant range. When $d$ is relaxed to $0.5$, costs increase as expected under a looser chance constraint. Even then, the increase is controlled, does not result in catastrophic cost, and reward remains competitive. This behavior matches the role of $d$: lower values allow the policy to prioritize reward at the expense of constraint-satisfaction, while higher values enforce stricter constraint satisfaction.

Overall, these results demonstrate that **TraCeS** is robust to the choice of $d$ across a practical range. Our findings indicate little sensitivity for $d \geq 0.8$, except under extremely relaxed desired acceptability rates.

*Table 12.* Sensitivity of **TraCeS** to the desired acceptability rate hyperparameter $d$ (evaluation environment) in the low-data regime (number of labeled trajectories is fixed to 1000 for all runs), with $d$ varied from 0.99 to 0.5. Non-violating policies are shown in black bold; the best among non-violating methods is highlighted in blue. Violating policies are shown in gray. Standard deviations shown in (). As $d$ decreases, cost increases, reflecting looser chance constraints, while both constraint-satisfaction and reward remain robust for reasonable $d$ values ($\geq 0.8$). See Appendix C.10 for further discussion.

| Task | Metric | TraCeS ($d = 0.99$) | TraCeS ($d = 0.9$) | TraCeS ($d = 0.8$) | TraCeS ($d = 0.5$) |
|---|---|---|---|---|---|
| Ant | Reward | **2506.5 (188.9)** | **2522.1 (150.1)** | **2377.8 (260.7)** | 2540.2 (205.6) |
| | Cost | **20.4 (7.5)** | **16.3 (4.7)** | **20.5 (3.9)** | 27.3 (8.4) |
| | Labeled Trajectories | **1000 (0)** | **1000 (0)** | **1000 (0)** | 1000 (0) |
| Hopper | Reward | **1243.5 (165.5)** | **1426.3 (122.7)** | 939.6 (1060.2) | 1173.8 (402.9) |
| | Cost | **18.0 (9.3)** | **18.0 (4.7)** | 26.7 (12.4) | 49.1 (27.4) |
| | Labeled Trajectories | **1000 (0)** | **1000 (0)** | 1000 (0) | 1000 (0) |
| AntRun | Reward | **460.9 (53.2)** | **494.0 (63.4)** | **476.4 (34.4)** | 498.5 (22.4) |
| | Cost | **19.6 (2.7)** | **23.0 (4.0)** | **24.1 (1.7)** | 30.5 (5.1) |
| | Labeled Trajectories | **1000 (0)** | **1000 (0)** | **1000 (0)** | 1000 (0) |
| CarRun | Reward | **557.0 (5.6)** | **566.4 (5.8)** | **546.6 (41.1)** | 545.9 (64.0) |
| | Cost | **20.0 (3.7)** | **21.4 (4.8)** | **24.1 (11.9)** | 41.0 (53.1) |
| | Labeled Trajectories | **1000 (0)** | **1000 (0)** | **1000 (0)** | 1000 (0) |

## C.11. Sensitivity Analysis of Labeling Noise

We evaluate the robustness of **TraCeS** to labeling noise, where each partial trajectory segment may be mislabeled with a given probability (i.e., a non-violating trajectory may be labeled as violating, and vice versa). This setup simulates practical scenarios where human or automated feedback is imperfect. To study the effects of increasingly challenging annotation conditions, we inject mislabeling at rates from 0% (no noise) up to 20% (very high noise) during both offline pretraining and online continual refinement of the **TraCeS** violation estimator, while fixing the number of labeled trajectories to 1000 for all runs. Table 13 summarizes the results across four representative tasks.

Across low to moderate noise rates (up to 10%), **TraCeS** maintains strong constraint-satisfaction performance: the cost metric remains near or below the unknown threshold in all tasks, and reward declines only slightly. This suggests a conservative bias in constraint estimation under uncertainty (see Remark 4.6 for a theoretical explanation based on the Jensen's gap). At higher noise levels (20%), a modest increase in cost is observed in some tasks, but cost remains close to the unknown threshold without catastrophic cost increase. Reward declines gradually as noise increases, consistent with the **TraCeS** estimator becoming more conservative in order to remain constraint-satisfying as uncertainty rises (see Remark 4.6).

Overall, these results demonstrate that **TraCeS** is robust to moderate levels of annotation noise, maintaining strong constraint-satisfaction and competitive reward even when label information is partially corrupted. At very high noise rates (20%), some degradation is unavoidable; however, the method typically becomes more conservative. Noisy supervision increases the variance of predicted acceptability $\hat{p}_\tau$, which widens the Jensen gap (Theorem 4.5) and makes the Jensen-based surrogate constraint a weaker (more conservative) lower bound. As a result, **TraCeS** may adopt stricter behavior to maintain feasibility, trading off some reward — an often desirable property when feedback is noisy or inconsistent.

*Table 13.* Sensitivity of **TraCeS** to label noise in the low-data regime (1000 labeled trajectories per run), with mislabeling rate ranging from 0% to 20%. Non-violating policies are shown in black bold; the best among non-violating methods is highlighted in blue. Violating policies are shown in gray. Standard deviations shown in (). As label noise increases, reward declines slightly due to more conservative estimation (see Theorem 4.5). Cost remains near or below the unknown threshold for all but the highest noise rates, where a slight increase can occur—an expected limitation under extremely noisy labels. Overall, **TraCeS** maintains robust constraint-satisfaction and reward performance in the presence of moderate label noise. See Appendix C.11 for further discussion.

| Task | Metric | TraCeS (0% noise) | TraCeS (5% noise) | TraCeS (10% noise) | TraCeS (20% noise) |
|------|--------|-------------------|-------------------|--------------------|--------------------|
| Ant | Reward | **2522.1 (150.1)** | **2606.7 (217.5)** | **2292.5 (433.6)** | **2346.3 (375.3)** |
| | Cost | **16.3 (4.7)** | **15.9 (7.8)** | **24.4 (14.1)** | **17.2 (18.9)** |
| | Labeled Trajectories | **1000 (0)** | **1000 (0)** | **1000 (0)** | **1000 (0)** |
| Hopper | Reward | **1426.3 (122.7)** | 1343.9 (90.2) | 1372.5 (119.1) | 1309.6 (154.9) |
| | Cost | **18.0 (4.7)** | 33.9 (20.6) | 27.3 (29.2) | 49.0 (55.2) |
| | Labeled Trajectories | **1000 (0)** | 1000 (0) | 1000 (0) | 1000 (0) |
| AntRun | Reward | **494.0 (63.4)** | **468.6 (43.1)** | **363.6 (105.0)** | 310.3 (89.7) |
| | Cost | **23.0 (4.0)** | **21.0 (4.0)** | **18.8 (8.4)** | 25.8 (15.0) |
| | Labeled Trajectories | **1000 (0)** | 1000 (0) | 1000 (0) | 1000 (0) |
| CarRun | Reward | **566.4 (5.8)** | 544.3 (40.2) | 532.0 (45.7) | 456.8 (149.2) |
| | Cost | **21.4 (4.8)** | 17.8 (3.1) | 11.7 (6.7) | 27.4 (35.7) |
| | Labeled Trajectories | **1000 (0)** | 1000 (0) | 1000 (0) | 1000 (0) |

## C.12. Sensitivity to Log-Credit Clamping

Our estimator outputs per-step log-credits $\log \hat{P}_t^\Delta \leq 0$, which are summed over time to form a trajectory-level log non-violation score. For numerical stability (especially in long-horizon settings where many small probabilities are multiplied), we impose a lower bound (clamp) on the per-step log-credits:

$$\log \hat{P}_t^\Delta \leftarrow \max(\log \hat{P}_t^\Delta, c),$$

where $c < 0$ (e.g., $c = -7$) prevents extreme negative values that can lead to underflow and unstable gradients.

Table 14 reports a sensitivity study comparing $c = -7$ and a looser clamp $c = -10$ in a low-data regime (1000 labeled trajectories per run). We chose $c = -7$ as a conservative default and $c = -10$ as a stress test allowing rarer, more extreme per-step penalties.

Across tasks, **TraCeS** remains non-violating for both clamp values, with performance differences that are modest and task-dependent (neither clamp dominates uniformly). This suggests that clamping primarily serves as a stability device rather than a tuning knob critical to the qualitative behavior of **TraCeS**.

*Table 14.* Sensitivity of **TraCeS** to the lower-bound clamp on per-step log-credits, $\log \hat{P}_t^\Delta \geq c$, in the low-data regime (1000 labeled trajectories per run). Clamping is used for numerical stability when summing log-credits over long horizons. Non-violating policies are shown in black bold; the best among non-violating methods is highlighted in blue. Violating policies are shown in gray. Standard deviations shown in ().

| Task | Metric | TraCeS ($c = -7$) | TraCeS ($c = -10$) |
|------|--------|-------------------|---------------------|
| Ant | Reward | **2522.1 (150.1)** | **2917.8 (162.9)** |
| | Cost | **16.3 (4.7)** | **16.3 (4.0)** |
| | Labeled Trajectories | **1000 (0)** | **1000 (0)** |
| Hopper | Reward | **1426.3 (122.7)** | 1350.3 (365.2) |
| | Cost | **18.0 (4.7)** | 21.2 (7.1) |
| | Labeled Trajectories | **1000 (0)** | 1000 (0) |
| AntRun | Reward | **494.0 (63.4)** | **538.5 (14.7)** |
| | Cost | **23.0 (4.0)** | **22.0 (1.4)** |
| | Labeled Trajectories | **1000 (0)** | **1000 (0)** |
| CarRun | Reward | **566.4 (5.8)** | 551.1 (44.9) |
| | Cost | **21.4 (4.8)** | 25.0 (3.8) |
| | Labeled Trajectories | **1000 (0)** | 1000 (0) |

## C.13. Sensitivity to Checkpoint Interval (Labeling Sparsity)

**TraCeS** can be trained from sparse prefix labels ("checkpoints") extracted from labeled rollouts. A smaller checkpoint interval provides denser supervision along each trajectory, while a larger interval makes feedback sparser. To study this effect, we vary the checkpoint interval while keeping the overall labeling budget fixed at 1000 labeled trajectories per run.

Table 15 shows that **TraCeS** remains broadly robust as supervision becomes sparser, though safety and reward can degrade on some tasks. Intuitively, increasing the checkpoint interval reduces the number of supervised prefixes available for credit assignment, making it harder for the estimator to (i) localize the timesteps responsible for a sharp drop in predicted non-violation probability and (ii) distinguish "incidental" behaviors that co-occur with violations from behaviors that are diagnostic of violation. With fewer labeled prefixes, the estimator sees fewer "before/after" comparisons along the same rollout. As a result, behaviors that appear inside violating trajectories have fewer nearby non-violating prefixes to serve as contrast, making it harder to tell which timesteps are merely co-occurring and which are truly diagnostic of violation. This can degrade credit localization and, downstream, lead to either slightly higher violation rates or overly conservative policies.

This trend is most pronounced on Walker2d, where sparser supervision reduces reward substantially and can slightly compromise constraint satisfaction. A plausible explanation is that performant Walker2d behavior often operates close to the safety boundary in common safety benchmark instantiations; thus, small attribution errors near the boundary translate into meaningful policy changes. When the estimator is less certain about which portions of the rollout are truly responsible for violations, the policy improvement step can become conservative, trading off reward to maintain acceptability. In contrast, tasks such as Ant and HalfCheetah appear less sensitive in this regime, suggesting that either (a) violations are driven by more easily identifiable, coarse-grained failure modes, or (b) there is more slack between high-reward behavior and the safety boundary under the benchmark constraint.

*Table 15.* Sensitivity of **TraCeS** to checkpoint interval (labeling sparsity) in the low-data regime (1000 labeled trajectories per run). A checkpoint interval of $k$ means we supervise prefixes every $k$ environment steps within a labeled rollout (plus the terminal label). Non-violating policies are shown in black bold; the best among non-violating settings is highlighted in blue. Violating policies are shown in gray. Standard deviations shown in ().

| Task | Metric | TraCeS (20 steps) | TraCeS (40 steps) | TraCeS (100 steps) |
|---|---|---|---|---|
| Ant | Reward | 2522.1 (150.1) | 2766.8 (123.5) | 2803.1 (99.3) |
| | Cost | 16.3 (4.7) | 14.3 (3.6) | 14.0 (3.0) |
| | Labeled Trajectories | 1000 (0) | 1000 (0) | 1000 (0) |
| HalfCheetah | Reward | 2085.9 (422.9) | 2131.2 (280.3) | 1990.8 (257.3) |
| | Cost | 21.3 (2.6) | 28.3 (8.6) | 22.6 (7.8) |
| | Labeled Trajectories | 1000 (0) | 1000 (0) | 1000 (0) |
| Hopper | Reward | 1426.3 (122.7) | 1505.0 (161.4) | 1502.0 (125.4) |
| | Cost | 18.0 (4.7) | 24.5 (13.3) | 26.3 (20.2) |
| | Labeled Trajectories | 1000 (0) | 1000 (0) | 1000 (0) |
| Walker2d | Reward | 1656.9 (277.0) | 1398.2 (138.5) | 1380.3 (321.0) |
| | Cost | 22.6 (8.1) | 29.6 (12.5) | 28.0 (10.5) |
| | Labeled Trajectories | 1000 (0) | 1000 (0) | 1000 (0) |

## C.14. Terminal-Only Labels

Our main experiments use sparse checkpoint/prefix labels along each trajectory rather than dense per-step labels. Terminal-only supervision is a stricter special case in which only the final trajectory label is observed. This removes intermediate localization signal and makes long-horizon credit assignment substantially harder.

Table 16 reports results using a single end-of-episode label and 5K labeled trajectories on four representative tasks. The results are generally weaker than the sparse-checkpoint setting, as expected, but show that terminal-only labels can still provide usable supervision on several tasks. Thus, terminal-only supervision is supported by the formulation, while sparse checkpoint labels provide a more practical balance between labeling cost and credit-assignment difficulty.

*Table 16.* Performance of **TraCeS** with terminal-only labels using 5K end-of-episode labeled trajectories. Results are reported in the evaluation environment. Standard deviations are shown in parentheses.

| Task | Reward | Cost |
|------|--------|------|
| Ant | **2518.6 (210.1)** | **16.6 (6.7)** |
| HalfCheetah | 1530.4 (156.2) | 50.8 (51.6) |
| AntRun | 518.1 (20.7) | 26.6 (6.3) |
| CarRun | **511.2 (18.1)** | **6.2 (8.0)** |

## C.15. Additional Threshold Setting: $b = 50$

We additionally evaluate **TraCeS** under a different ground-truth cumulative-cost threshold, $b = 50$, to test robustness of the learned violation-credit pipeline to the underlying acceptability criterion. As in our main experiments, the agent does not observe the oracle cost function or threshold during training; we use the oracle only to *construct* binary accept/reject labels for benchmarking. Table 17 reports evaluation performance. Concretely, changing $b$ changes the induced labeling rule $\Psi(\tau) = \mathbb{1}\{\sum_t C(s_t, a_t) \leq b\}$; this experiment checks whether **TraCeS** remains stable under such shifts in $\Psi$.

Across tasks, **TraCeS** remains constraint-satisfying and achieves competitive reward under this alternative threshold. In particular, **TraCeS** is non-violating on all supported tasks in Table 17 and achieves higher reward than the *C-T* baseline on Ant and Hopper, while matching it closely on CarRun. PPO-Lagrangian (oracle) provides an upper bound when full access to the true cost signal and threshold is available, whereas RLSF is not supported on some tasks in our suite. Overall, these results indicate that **TraCeS** remains effective when the underlying acceptance boundary changes, consistent with our formulation that optimizes a chance-style acceptability constraint induced by the (unknown) labeling rule.

*Table 17.* Performance (eval environment) for ground-truth threshold $b = 50$. Non-violating policies are shown in black bold; the best among non-violating methods is highlighted in blue. Violating policies are shown in gray. Standard deviations shown in ().

| Task | Metric | PPO-Lagrangian (Oracle) | RLSF (Partial Knowledge) | *C-T* Baseline (Zero Knowledge) | **TraCeS (Ours)** (Zero Knowledge) |
|------|--------|------------------------|--------------------------|-------------------------------|------------------------------------|
| | Reward | **3238.8 (53.5)** | **1737.4 (477.3)** | 3177.8 (90.3) | **2523.8 (244.9)** |
| Ant | Cost | **28.5 (19.3)** | **4.4 (6.7)** | 56.3 (29.8) | **38.9 (8.2)** |
| | Labeled Trajectories | **NA** | **1175 (274)** | 5000 (0) | **5000 (0)** |
| | Reward | **1384.6 (488.8)** | 1553.0 (99.5) | 1324.5 (658.9) | **1372.9 (231.0)** |
| Hopper | Cost | **29.5 (9.1)** | 52.7 (71.0) | 408.0 (329.0) | **47.9 (12.4)** |
| | Labeled Trajectories | **NA** | 4365 (1218) | 5000 (0) | **5000 (0)** |
| | Reward | **696.1 (20.0)** | Non- | 658.2 (20.7) | **511.5 (22.2)** |
| AntRun | Cost | **39.0 (27.8)** | supported | 56.5 (22.8) | **43.9 (3.1)** |
| | Labeled Trajectories | **NA** | task | 5000 (0) | **5000 (0)** |
| | Reward | 599.2 (266.0) | Non- | **649.7 (5.3)** | **641.4 (14.4)** |
| CarRun | Cost | 60.7 (55.0) | supported | **49.5 (1.4)** | **49.9 (1.8)** |
| | Labeled Trajectories | NA | task | **5000 (0)** | **5000 (0)** |

## D. TraCeS Pseudo Code

We provide the pseudo-code of **TraCeS** training loop in Algorithm 1, which iteratively trains the violation estimator and updates the policy via constrained optimization.

---

**Algorithm 1 TraCeS Training Loop.** This pseudocode describes the full **TraCeS** training loop, including selective annotation, violation estimator learning, and constrained policy optimization.

---

1: Initialize policy $\pi_\theta$, critic networks, violation estimator $f_w$, and data buffers: labeled $\mathcal{D}$ and temporary $\mathcal{B}$
2: Optionally pretrain $f_w$ if labeled trajectories exist in $\mathcal{D}$
3: **for** each training iteration **do**
4:     Roll out trajectories using $\pi_\theta$ and store in $\mathcal{B}$
5:     **if** number of trajectories in $\mathcal{B} \geq$ maximum number allowed **then**
6:         Compute coefficient of variation (CV) scores for trajectories in $\mathcal{B}$ (see Section 4.3)
7:         Select subset with highest CV and query binary violation labels
8:         Add labeled trajectories to $\mathcal{D}$, and clear $\mathcal{B}$
9:     **end if**
10:     Update violation estimator $f_w$ using maximum likelihood estimation (MLE) on labeled $\mathcal{D}$
11:     Use $f_w$ to estimate per-step log-credits $\log \hat{P}_t^\Delta$ for current rollouts
12:     Update Lagrange multiplier and constraint critic using the estimated log-credits $\log \hat{P}_t^\Delta$
13:     Update reward critic using standard PPO-Lagrangian targets
14:     Update policy $\pi_\theta$ using PPO-Lagrangian objective
15: **end for**

---

## E. Limitations

**Constraint violations during training.** Like other Lagrangian-based approaches (Ha et al., 2021; Ray et al., 2019; Stooke et al., 2020), **TraCeS** optimizes for *eventual* constraint satisfaction rather than guaranteeing zero violations throughout training. In particular, early exploration can produce unsafe rollouts before the violation estimator and policy stabilize. Lagrangian updates can also be sensitive to multiplier initialization and step sizes; we mitigate this using standard settings from OmniSafe (Ji et al., 2024), but stability can still vary by environment. Techniques that provide stronger *training-time* safety guarantees (e.g., safety filters or conservative exploration) are complementary and orthogonal to our focus.

**Availability of labeled trajectories.** **TraCeS** requires trajectory- or prefix-level accept/reject labels. While this supervision is substantially cheaper than dense per-step costs, it still requires a labeling pipeline (human or automated monitor). In many domains, such labels can be collected retrospectively after deployment (e.g., when failures are identified) to incrementally refine the estimator and policy, or obtained in simulation prior to deployment to reduce physical risk.

**Consistency and stationarity of the safety signal.** Our formulation assumes the labeling rule $\Psi$ is reasonably consistent over the training horizon. In practice, supervision may be noisy or non-stationary (e.g., shifting tolerance as operators become stricter). We empirically study robustness to label noise and changes in the induced acceptability boundary (Appendix C.11, Appendix C.15), but highly inconsistent or drifting supervision may require additional calibration or adaptation mechanisms.

