# OpenReview forum: "TraCeS: Learning Per-Timestep Constraint-Violation Credit from Sparse Trajectory-Level Labels"
_ICML.cc/2026/Conference — ICML 2026 regular_

### Official Review · Reviewer_Ey61 · 2026-02-24

**Soundness:** 3
**Presentation:** 3
**Significance:** 3
**Originality:** 3
**Overall Recommendation:** 4
**Confidence:** 4

**Summary:**

The authors addresses a safe reinforcement learning problem where safety constraints are implicit and cannot be densely measured. Here,  supervision is limited to coarse approvals or rejections of whole trajectories. The authors propose Trajectory-based Constraint Estimation for Safety (TraCeS) for learning per-timestep violation credit from such sparse trajectory-level labels. TraCeS trains a sequential violation estimator whose per-step credits factorize the predicted probability that a trajectory has not yet violated the constraint. The authors provide a theoretical analysis of the approximation gap introduced by the learning objective, and demonstrate empirically that TraCeS improves constraint satisfaction and feedback efficiency over baselines across multiple continuous-control benchmarks.

**Compliance With Llm Reviewing Policy:**

Affirmed.

**Final Justification:**

This is a good paper with solid algorithmic and empirical contributions.

**Key Questions For Authors:**

1. Could you elaborate more on all the assumptions behind theoretical restuls?

**Limitations:**

I think the authors have adequately discussed the limitations in Appendix E.

**Strengths And Weaknesses:**

### Strengths

1. This paper is well-written and easy to follow. The problem settings are motivated and useful in many applications.

1. The proposed method is technically sound and clearly explained with rigor mathematics.

1. The empirical experiments are sufficiently conducted with many standard benchmark tasks and baseline algorithms.

1. The source code is attached and I found it is easy to reproduce the experimental results.

### Weaknesses

1. When releasing the source code, please consider it refactoring (since I feel it is a little bit messy).

1. If I understand correctly, this paper does not clearly state all the assumption behind theoretical analyses. I would like the authors to add mathematical descriptions regarding assumptions rather than texts.

1. (Minor) I am wondering whether this approach can be used in offline safe RL paradigm given the first limitation (i.e., constraint violations during training)

---

> ### Author Rebuttal · Authors · 2026-03-31
>
> Thank you for the helpful review and positive assessment. We are glad the motivation, method, and empirical evaluation came across clearly.
>
> **On the theoretical assumptions.**
>
> We agree these assumptions can be surfaced more explicitly. In the current version, the main assumptions behind the theoretical development are located at:
>
> 1. Monotone / irreversible violation structure (formalized as Assumption 3.1): once a prefix is violating, later prefixes remain violating. This is the key structural assumption underlying the “not-yet-violated” formulation and the monotonicity reflected in Eq. (4).
>
> 2. Standard realizability/capacity assumption in Lemma 4.4: if the estimator class is expressive enough to represent the relevant accept/reject rule and is fit sufficiently well on the on-policy trajectory distribution, then the learned trajectory-level score $\hat p_\tau$ is a good proxy for the acceptability indicator on that distribution.
>
> 3. Clamping-induced lower bound in Theorem 4.5: clamping $\log \hat{P}_{t}^{\Delta} \ge -7$ (Section 4.1.2) ensures a positive lower bound on trajectory-level non-violation scores, which yields the $\epsilon>0$ condition used in Theorem 4.5.
>
> We will revise the paper to surface these assumptions more explicitly.
>
> **On offline safe RL.**
>
> Yes, the framework is in principle compatible with an offline setting, since the violation estimator can also be trained on offline trajectory data. In fact, this particular limitation (constraint violations during training) is less restrictive in offline RL, because any constraint violations are confined to the pre-collected dataset rather than arising during interactive training.
>
> However, this paper focuses on the online setting because iterative relabeling and trajectory selection are central there, while current offline safe RL benchmarks usually assume known costs rather than implicit accept/reject supervision. We agree this is a natural extension.
>
> Thank you also for the suggestion on code organization; we will clean up the public release.

---

> > ### Author Rebuttal · Reviewer_Ey61 · 2026-04-01
> >
> > Thank you for the careful responses. Given the revised manuscript cannot be submitted at ICML, I will keep my score. I hope the authors to surface these assumptions more explicitly as they promise.

---

> > > ### Author Response · Authors · 2026-04-04
> > >
> > > Thank you for your time and feedback. We will definitely surface the theoretical assumptions more explicitly in the final manuscript.

---

### Official Review · Reviewer_CJkW · 2026-03-11

**Soundness:** 3
**Presentation:** 3
**Significance:** 3
**Originality:** 3
**Overall Recommendation:** 5
**Confidence:** 4

**Summary:**

TraCeS proposes a framework for learning safe policies under unknown cost functions and thresholds, utilizing sparse trajectory-level labels. Its core innovation lies in decomposing the trajectory-level non-violation probability into the product of per-step credits and transforming it, via Jensen's inequality, into a standard Constrained Markov Decision Process (CMDP) optimization problem. Additionally, the algorithm employs the coefficient of variation (CV) to prioritize the most informative trajectories for labeling, thereby enhancing feedback efficiency.

**Compliance With Llm Reviewing Policy:**

Affirmed.

**Final Justification:**

After considering the paper and the authors’ rebuttal, I maintain my positive assessment and support acceptance. I find the work technically solid, well motivated, and clearly presented. The paper addresses an important problem in safe policy learning under sparse feedback, and the proposed formulation is both original and practically meaningful. Overall, I view this as a strong and worthwhile contribution to the area.

**Key Questions For Authors:**

1. Theorem 4.5 provides a theoretical bound on the Jensen gap, but the constant $\frac{1}{\epsilon^2}$ depends on the minimum trajectory non-violation probability. Given the clamping value $c=-7$ and a horizon $T=1000$, $\epsilon$ can be extremely small, making the bound vacuous. Could the authors discuss the tightness of this bound in practical long-horizon scenarios?
2. The trajectory selection strategy relies on the coefficient of variation (CV) derived from the estimator's sampling noise (aleatoric uncertainty). However, active learning typically benefits more from epistemic uncertainty (e.g., via model ensembles). Why was epistemic uncertainty ignored, and does this lead to sampling bias in regions where the estimator is 'confidently wrong'?
3. The paper claims to learn from 'sparse trajectory-level labels,' yet the experiments use prefix labels every 5 to 20 timesteps. This is significantly denser than a single label per episode. How does TraCeS perform under truly sparse conditions where only one terminal label is provided? The sensitivity analysis in C.12 suggests a performance drop as sparsity increases; please clarify the 'sparse' claim.
4. The violation estimator is pre-trained to an initial accuracy of 89%-97% before policy optimization begins. Does the success of the iterative loop and CV selection depend on this high-quality initialization? It would be insightful to see the performance of TraCeS when starting from a randomly initialized estimator.

**Limitations:**

yes

**Strengths And Weaknesses:**

1. Soundness
The proposed framework is technically rigorous, utilizing a Jensen lower bound to transform trajectory-level probability constraints into a tractable per-step surrogate cost . While the method is validated across 12 diverse continuous-control tasks, the theoretical approximation gap defined in the paper may become loose in long-horizon settings due to the necessary numerical clamping of log-credits . Additionally, the empirical use of checkpoint labeling every 5 to 20 steps slightly deviates from the strict "trajectory-level" sparse feedback claim emphasized in the title.
2. Presentation
The paper is well-structured, with precise language and a clear logical flow that makes the overall narrative very easy to follow.
3. Significance
This work addresses the vital problem of safe policy learning under implicit constraints where both the cost function and threshold remain unobserved. By demonstrating superior label efficiency compared to zero-knowledge baselines and maintaining robustness under moderate annotation noise or human feedback, the framework offers significant utility for real-world applications like autonomous driving . It successfully provides a practical bridge between sparse human judgments and formal constrained optimization.
4. Originality
The primary innovation lies in the creative adaptation of multiplicative decomposition logic, traditionally found in survival analysis, to model irreversible safety violations as a sequence of per-step violation credits. Furthermore, the integration of a selective labeling strategy based on the coefficient of variation (CV) presents a novel approach to addressing distribution shifts and minimizing annotation burdens in Safe RL. This combination of existing probabilistic tools with constrained policy optimization offers a fresh perspective on credit assignment.

---

> ### Author Rebuttal · Authors · 2026-03-31
>
> Thank you for the careful and very helpful review. We are glad the core formulation and presentation came across clearly.
>
> **1. Tightness of Theorem 4.5 in long horizons.**
>
> We agree that Theorem 4.5 is a worst-case bound and can become loose in long-horizon settings because the trajectory-level lower bound $\epsilon$ may be very small after composing many clamped per-step credits. Our intent was not to claim a numerically tight certificate in all such regimes, but to show that the Jensen gap scales with $\mathrm{Var}(\hat p_\tau)$ and therefore shrinks as the estimator becomes more stable on the on-policy trajectory distribution.
>
> Empirically, TraCeS remains robust on horizon-1000 tasks and in robustness studies (e.g., Appendix C.10-C.11). Importantly, looseness here tends to make the surrogate constraint _more conservative_, rather than overly optimistic, which is acceptable in our setting.
>
> **2. CV-based selection and epistemic uncertainty.**
>
> We agree that our current CV score is not a full ensemble-style epistemic uncertainty estimator. It is better viewed as a lightweight uncertainty proxy based on the estimator’s predicted per-step sampling dispersion, aggregated at the trajectory level. We chose it because it is simple, computationally cheap, and effective in the online selective-labeling loop; the trajectory-selection ablation (Appendix C.7) shows a gain over random selection, especially on more challenging tasks.
>
> In the current online setting, the risk of “confidently wrong” regions is partly mitigated by repeatedly retraining the estimator on newly collected labeled trajectories as the policy distribution shifts, so such errors are not frozen indefinitely. We also agree that an ensemble-based epistemic measure is a promising extension, especially for further reducing this risk under distribution shift, but we do not rely on that stronger claim in the current paper.
>
> **3. Clarifying the “sparse trajectory-level” claim.**
>
> We clarify that our main experiments use sparse checkpoint/prefix labels every 5-20 steps (with Appendix C.12 studying sensitivity to checkpoint interval), rather than dense per-step labels. Terminal-only supervision is a stricter special case of the same formulation. To clarify this, we ran a new **terminal-only experiment** on 4 representative tasks using a single end-of-episode label and 5K labeled trajectories:
> | Task | Reward | Cost |
> |---|---:|---:|
> | Ant | 2518.6 (210.1) | 16.6 (6.7) |
> | HalfCheetah | 1530.4 (156.2) | 50.8 (51.6) |
> | AntRun | 518.1 (20.7) | 26.6 (6.3) |
> | CarRun | 511.2 (18.1) | 6.2 (8.0) |
>
> As expected, these results are reasonable but weaker than the sparse-checkpoint setting reported in the paper, because terminal-only labels remove intermediate localization signal and make long-horizon credit assignment substantially harder. However, they show that **terminal-only labels are a supported but harder special case**, rather than a different formulation. We will revise the wording to make this distinction explicit.
>
> **4. Dependence on high-quality pretraining.**
>
> We also ran a new **random-initialization experiment** (same 4 tasks). TraCeS still learns viable policies without offline pretraining:
> | Task | Reward | Cost |
> |---|---:|---:|
> | Ant | 2655.3 (201.0) | 17.7 (2.7) |
> | HalfCheetah | 1934.5 (439.2) | 22.9 (2.7) |
> | AntRun | 500.3 (35.9) | 23.5 (1.9) |
> | CarRun | 559.0 (17.1) | 22.3 (3.4) |
>
> This suggests that pretraining improves sample efficiency/stability, but the iterative estimator-policy loop **does not fundamentally depend on near-perfect initialization**.
>
> We appreciate these suggestions and will incorporate the above clarifications in the revision.

---

> > ### Author Rebuttal · Reviewer_CJkW · 2026-04-02
> >
> > The authors have addressed my concerns. Accordingly, I maintain my positive recommendation.

---

> > > ### Author Response · Authors · 2026-04-04
> > >
> > > Thank you for reviewing our rebuttal. We greatly appreciate your time and helpful suggestions.

---

### Official Review · Reviewer_HK7n · 2026-03-12

**Soundness:** 3
**Presentation:** 3
**Significance:** 3
**Originality:** 3
**Overall Recommendation:** 4
**Confidence:** 3

**Summary:**

This paper focuses on safe RL and proposes a new problem formulation for constrained MDPs, where the cost for each timestep and the threshold for the task cost are not observable. Specifically, the authors introduce TraCeS, which predicts step-wise violation credits from binary trajectory-level cost labels. The method is evaluated on 12 tasks and demonstrates significant performance improvements.

**Compliance With Llm Reviewing Policy:**

Affirmed.

**Final Justification:**

I would like to keep my positive score.

**Key Questions For Authors:**

I assume the framework is similar to return decomposition to some extent, where continuous episodic rewards are typically decomposed into time-wise rewards. However, this appears to be an ill-posed problem because information is lost when a set of rewards/costs is scalarized into a single episodic value. The situation seems even more challenging in the case studied in this paper, where the observed signals are binary. How do the authors address this issue? Also, I would suggest including the return decomposition method as related work.

Could the authors provide an example illustrating how the timestamp of a violation can be unknown while labeling a subsequence of a trajectory? Is this scenario practical? Or is it only practical under the condition that a violation occurs and then persists, as assumed in Assumption 3.1? If the assumption does not hold, is the proposed method and setting still practical? If not, how could the proposed method be extended to handle such cases?

How many timesteps are there in a single trajectory? Is the method robust to longer trajectories? The gradient seems to vanish as the trajectory becomes longer.

Additionally, due to the newly introduced predictor, how is the increased computational cost and scalability of this method?

**Limitations:**

Yes

**Strengths And Weaknesses:**

The paper is well written and easy to follow. The problem setting is important and practically relevant, and the proposed formulation is well motivated, especially since in many realistic scenarios the cost function and safety threshold are not directly observable.

However, while the authors avoid requiring a cumulative cost threshold, they introduce a desired acceptability rate as a replacement. As a result, it is not entirely clear whether this fully removes the need for manually specified safety-related hyperparameters. Moreover, the paper does not clearly quantify the additional computational cost introduced by the proposed method.

---

> ### Author Rebuttal · Authors · 2026-03-31
>
> Thank you for the thoughtful review and constructive questions. We are glad the motivation and problem setting came across clearly.
>
> **On the desired acceptability rate $d$.**
>
> We will clarify that $d$ is not a replacement for the unknown threshold $b$. The unknown violation rule and threshold are already encoded implicitly in the accept/reject labels; $d$ only specifies how often the learned policy should satisfy that latent rule. This is easier to interpret in practice, since choosing a target satisfaction rate in $[0,1]$ is typically much simpler than specifying an unknown cost structure and threshold directly. Appendix C.9 also shows robustness over a range of $d$ values.
>
> **Q1. Return decomposition.**
>
> We agree there is a conceptual connection to delayed credit-assignment methods such as return decomposition (e.g., RUDDER / RRD), and will add this discussion in related work.
>
> The key difference is that TraCeS does not attempt to uniquely recover the true oracle per-step cost from a scalar endpoint signal. Instead, it learns a _surrogate per-step violation credit_ that predicts the observed binary not-yet-violated supervision and supports constrained policy optimization. This is still underdetermined in the strict identifiability sense, but our setting exploits useful structure to make the problem tractable: (i) monotone “not-yet-violated” supervision structure, and (ii) sparse labeled prefixes/checkpoints rather than only a single scalar terminal outcome. The learned credit should therefore be viewed as a supervision-aligned surrogate for control, not as exact recovery of the hidden ground-truth cost.
>
> **Q2. Unknown violation timestamp & Assumption 3.1.**
>
> A labeled prefix only tells us whether violation has occurred by that point, not the exact first violating timestep inside the prefix. For example, if the prefix up to step 20 is non-violating and the prefix up to step 40 is violating, then the first violation occurred somewhere between steps 21-40, but its exact timestamp is still unknown. This is practical when labels come from sparse human/monitor checkpoints or retrospective review rather than dense per-step annotation: an annotator may be able to say “by this point the rollout is already unacceptable” without labeling the precise first violating transition, especially when acceptability depends on cumulative/contextual evidence rather than a single obvious event.
>
> Assumption 3.1 is the intended scope: monotone / irreversible violation settings, e.g., once a collision occurs, a restricted region is entered, or a cumulative budget is exceeded, later actions do not erase that earlier violation. This covers an important class of safety problems centered on preventing irreversible bad events (informally, “nothing bad happens”), which is the focus of the paper.
>
> If Assumption 3.1 does not hold, then the current formulation is no longer the right abstraction; handling such cases would likely require different machinery (e.g., richer latent event-state models or other structured machinery for properties where later behavior can change whether a trajectory is ultimately acceptable), which we view as an interesting but out-of-scope extension.
>
> **Q3. Long horizons.**
>
> The task horizons are 1000 (MuJoCo), 500 (Circle), and 100/200 (Run), as detailed in Appendix B.4. Empirically, TraCeS remains effective on the horizon-1000 tasks, so the method is not limited to short episodes.
>
> We mitigate long-horizon credit-assignment difficulty (and associated vanishing-gradient issues) through: (i) a recurrent summary model that compresses prefix information; (ii) sparse checkpoint/prefix labels along long trajectories rather than only a single terminal label; (iii) periodic retraining on newly labeled data as the policy distribution evolves; and (iv) operating in log space with clamping for numerical stability. These design choices are also why terminal-only supervision is a supported but harder special case (Appendix C.12; see also our response to reviewer CJkW, point 3).
>
> **Q4. Computational cost and scalability.**
>
> We clarify that the added estimator is lightweight (architecture details in Appendix B.4.3): per-step overhead is one extra estimator forward pass plus conditioning the policy / constraint critic on a small latent summary. The main extra cost comes from periodic estimator retraining rather than rollout generation, and this retraining frequency is adjustable. Empirically, the method scales across 12 tasks spanning different observation dimensions and horizons up to 1000 steps.
>
> We additionally **measured wall-clock overhead** at matched environment steps. On CarRun, TraCeS required 260.9s vs 249.9s for PPO-Lagrangian at 400K steps (+4.4%); on Ant (horizon 1000), TraCeS required 1573.0s vs 1512.4s at 1.0M steps (+4.0%). Most of the extra cost came from periodic estimator retraining rather than per-step inference.
>
> We appreciate these questions and will incorporate the above clarifications in the revision.

---

> > ### Author Rebuttal · Reviewer_HK7n · 2026-04-01
> >
> > Thanks for the authors' effort in address my concern. I would like to keep my score.

---

> > > ### Author Response · Authors · 2026-04-04
> > >
> > > Thank you for reviewing our rebuttal. We greatly appreciate your time and constructive feedback.

---

### Decision · Program_Chairs · 2026-04-30

**Decision:**

Accept (regular)

**Comment:**

This paper presents a method for safe reinforcement learning that learns to avoid bad actions using only simple "accept or reject" feedback instead of detailed cost functions. All reviewers agreed that the problem is highly practical and praised the proposed solution as creative, mathematically sound, and well-tested. The authors provided a strong rebuttal that successfully cleared up minor concerns regarding computational overhead, long-term tasks, and theoretical assumptions. Because the work is technically solid and all reviewers maintain a positive score, I recommend this paper for acceptance.